# Development of R7BP inhibitors through cross-linking coupled mass spectrometry and integrated modeling

Poorni R. Adikaram[1,4], Jian-Hua Zhang[1,4], Claire M. Kittock[1], Mritunjay Pandey[1], Sergio A. Hassan[2], Nicole G. Lue[1], Guanghui Wang[3], Marjan Gucek[3] & William F. Simonds [1]

Protein-protein interaction (PPI) networks are known to be valuable targets for therapeutic intervention; yet the development of PPI modulators as next-generation drugs to target specific vertices, edges, and hubs has been impeded by the lack of structural information of many of the proteins and complexes involved. Building on recent advancements in cross-linking mass spectrometry (XL-MS), we describe an effective approach to obtain relevant structural data on R7BP, a master regulator of itch sensation, and its interfaces with other proteins in its network. This approach integrates XL-MS with a variety of modeling techniques to successfully develop antibody inhibitors of the R7BP and RGS7/Gβ5 duplex interaction. Binding and inhibitory efficiency are studied by surface plasmon resonance spectroscopy and through an R7BP-derived dominant negative construct. This approach may have broader applications as a tool to facilitate the development of PPI modulators in the absence of crystal structures or when structural information is limited.

---

[1] Metabolic Diseases Branch, National Institute of Diabetes and Digestive and Kidney Diseases, Bldg. 10/Rm 8C-101, Bethesda, MD 20892, USA. [2] Center for Molecular Modeling, Center for Information Technology, Bldg. 12/Rm 2049, Bethesda, MD 20892, USA. [3] Proteomics Core, National Heart Lung and Blood Institute, National Institutes of Health, Bldg. 10/Rm 8C-103A, Bethesda, MD 20892, USA. [4]These authors contributed equally: Poorni R. Adikaram, Jian-Hua Zhang. Correspondence and requests for materials should be addressed to W.F.S. (email: bills@niddk.nih.gov)

R7BP (a.k.a. RGS7BP, or regulator of G protein signaling 7-binding protein) is a membrane anchor and critical binding partner of the R7-RGS/Gβ5 duplex that acts as a GTPase-activating protein (GAP) to accelerate the intrinsic GTPase activity of certain guanine-nucleotide-binding Gα proteins in the nervous system and other tissues[1–7]. Members of the R7 subfamily (R7-RGS; composed of RGS 6, 7, 9, and 11) of the regulator of G protein signaling (RGS) superfamily[8] forms tight heterodimers with Gβ5, a structurally divergent and evolutionarily conserved member of the G protein beta family[9]. Recently, it was discovered that R7BP is a master regulator of both acute and chronic itch, as R7bp knockout mice exhibited reduced scratching behavior in response to all types of pruritogens (itch-inducing agents) tested, yet were morphologically indistinguishable from their wild-type littermates[10]. Itch, especially chronic itch that is often associated with eczema, psoriasis, hepatic cholestasis, uremia, and certain neuropathies, is a significant global health burden due to its high prevalence. More than 20% of the general population will suffer one or more forms of pathologic itch in their lifetime[11,12]; there are about 7.5 million patients with psoriasis alone[13–15]. However, very few effective anti-chronic itch drugs are clinically available, and the use of many of them is limited by undesirable side effects[15]. The interactions between R7-RGS/Gβ5 duplexes and R7BP thus provide novel potential targets to block R7BP-dependent itch, allowing the development of novel specific anti-itch drugs with potentially fewer side effects[10].

Although PPI networks can serve as valuable targets for the discovery of next-generation drugs[16–18], the development of PPI modulators, e.g., inhibitors, is difficult because the process frequently depends on the availability of high-resolution structures of the interacting proteins or multiprotein complexes. Many of these proteins are difficult to crystalize[19,20], as is the case with R7BP. Further difficulty arises from the fact that R7BP shares no conserved protein domains with homologs for which structural information is already available. Consequently, we were compelled to search for alternative ways to understand its structural characteristics. Recent advances in integrative structural biology have made it possible to obtain key structural information on individual proteins and complexes, and gain insight into the relevant interactions in the network. These approaches use traditional X-ray and NMR spectroscopy and cryo-electron microscopy (cryo-EM) in combination with emerging techniques, such as computational modeling guided by distance restraints inferred from protein cross-linking coupled with mass spectrometry (XL-MS). XL-MS technology, where the spatial proximity of amino acid pairs is probed after treatment with suitable chemical cross-linkers[21–24], has been increasingly used to determine protein interaction sites because it reflects real-time intra-molecular and inter-molecular interactions[25,26]. The most significant and attractive advantages of XL-MS are simplicity, speed, and cost-effectiveness[26,27]. Its value has recently been demonstrated by the structural resolution of the human 26S proteasome using the disuccinimidyl sulfoxide (DSSO) cross-linker[27]. Although the structural information provided by XL-MS is sometimes limited due to the location, absence, or accessibility of linkable amino acid side chains, the deduced distance restraints nevertheless provide sufficient guiding information to refine the modeling process, so that experimentally validated insight into a protein structure and its interaction mechanisms can be produced[28–31]. The structural data obtained through this process could thus facilitate the rational design of protein-function or network-function modulators during drug development.

After reviewing and testing a number of different methods publicly available, we describe here a practical, yet robust approach that combines XL-MS and a selected set of modeling software, namely, the protein-structure prediction server I-TASSER[32], the protein–protein docking server ClusPro[33], and the academic version of the macromolecular simulation program CHARMM[34]. Data analysis suggests a "lobster-like" (or homarine) conformation of R7BP, containing an extensive head-to-tail-binding groove that interacts with the modeled RGS7/Gβ5 duplex, the crystal structure of which was recently reported[35]. We demonstrate the usefulness of this structural information by developing dominant negative and antibody inhibitors of R7BP functions, providing a promising first step to develop next-generation anti-itch drugs. This approach is effective and affordable and has the potential to facilitate the search for PPI inhibitors in the absence of relevant high-resolution structural information.

## Results

**Molecular modeling revealed a lobster-like R7BP structure.** R7BP acts as a membrane anchor for the R7-RGS/Gβ5 duplex to facilitate the GAP activity of the duplex directed against guanine nucleotide-binding Gα proteins[4,36]. The four members of the R7-RGS subfamily (RGS6, 7, 9, and 11) share extended sequence homologies (Supplementary Fig. 1), and their conformations can be modeled by I-TASSER (Supplementary Fig. 2). The recent discovery that R7BP is a master regulator of itch sensation[10] provides hope for the development of novel anti-itch therapeutics that might be effective towards multiple types of itch, especially for chronic itch, such as eczema and neuropathies, for which there are currently few specific and effective drugs whose use is not hampered by undesirable side effects. However, efforts to target R7BP function are presently hindered by the lack of knowledge of both its three-dimensional structure and the way it interacts with the R7-RGS/Gβ5 duplexes to form a functional heterotrimer. To overcome these difficulties, we modeled the R7BP structure through XL-MS calibrated molecular modeling using I-TASSER.

The reliability of I-TASSER was first assessed here by modeling the human versions of the R7BP-binding partners Gβ5 and RGS9, as the crystal structures of their mouse orthologs have been previously resolved[37]. Based on their extensive sequence homology, I-TASSER predictably generated three-dimensional structures for human Gβ5 and RGS9 comparable to their murine counterparts (Supplementary Fig. 2). We further confirmed the applicability of I-TASSER by modeling human RGS6, RGS7, and RGS11 which, along with RGS9, are members of the R7-RGS subfamily, and generated RGS9-like structures (Supplementary Fig. 2). These results indicated that I-TASSER could produce plausible three-dimensional models of proteins with similar sequences.

We then obtained a tentative model of the human R7BP structure also using this server. The resulting structure displayed a "lobster-like" conformation consisting of four major helices (H1–H4) and four short helices (sH1–sH4) (Fig. 1a, b). The four helices H1–H4 formed the lobster body, sH2 and sH3 formed the tail, sH1 and sH4 formed the claws, whereas the loops connecting sH1–H1, H2–H3, and H4–sH4 together formed the lobster head. This model was subsequently validated by the proximities between the specific cross-linked amino acids identified through XL-MS analysis of the purified unbound R7BP protein. A summary of all cross-linking experiments is provided in Supplementary Table 1, while detailed information on these experiments and results is provided in Supplementary Data 1. For R7BP-FL, 14 DSSO-linked pairs of lysine residues were identified as being in proximity to each other, as predicted in the model (Fig. 1c, Supplementary Table 2). This modeled structure is also consistent with previous models suggesting that R7BP contains a core of four helices

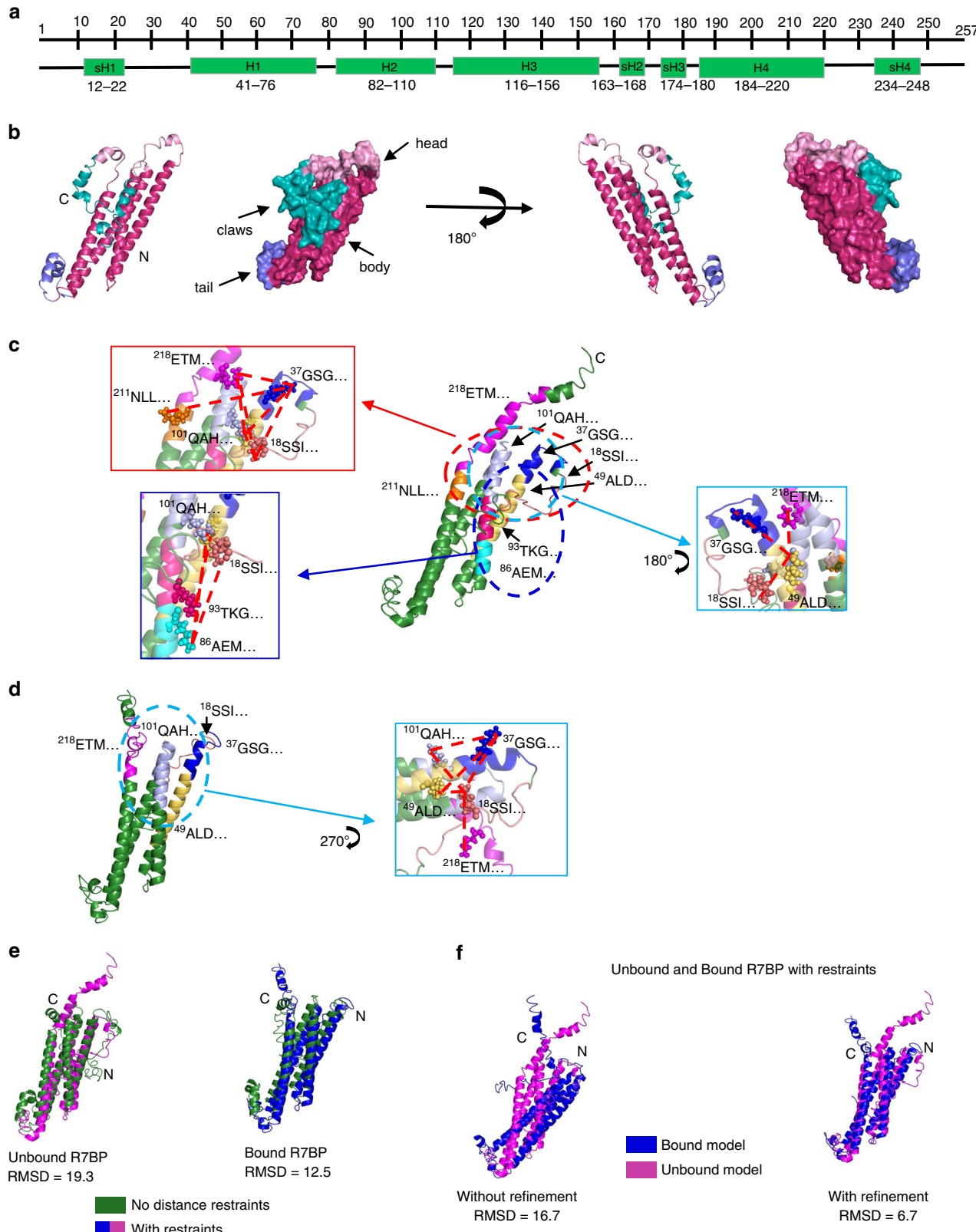

predicted to be critical for binding R7-RGS/Gβ5 duplexes[2]. We then refined the R7BP model using I-TASSER with the addition of distance restraints of 10 Å for the 14 lysine pairs (Fig. 1c). The XL-MS-calibrated model of the unbound R7BP was largely similar to the original model (Fig. 1b), except that in the refined model the

C-terminus protrudes out of the main body (Fig. 1c). This conformational change is consistent with the known role of the C-terminal region in the association of R7BP to the cell membrane, as it contains a polybasic domain and two vicinal cysteine residues that can undergo palmitoylation[3–5,38].

**Fig. 1** XL-MS-calibrated molecular modeling of R7BP. **a** Schematic of R7BP secondary structure as predicted by I-TASSER modeling. Four major helices (H1–H4) containing 30–40 amino acids each and four shorter helices (sH1–sH4) with 5–15 residues each were recognized. **b** I-TASSER modeling of the human R7BP protein showing the lobster-like 3D conformation in both cartoon and surface modes. The lobster head (pink), claws (teal), body (magenta), and tail (purple) are indicated with arrows. **c**, **d** Cartoon models of unbound (**c**) and RGS7/Gβ5-bound (**d**) R7BP with distance restraints are shown. Each peptide obtained through XL-MS is identified by the first three residues in its sequence. The insets depict a closer look at the cross-linked lysines (spheres) and their interactions (dashed lines). XL-MS identified 14 pairs of cross-linked intra-molecular peptides of the unbound and 6 pairs of the RGS7/Gβ5-bound R7BP protein. **e** Unbound and RGS7/Gβ5-bound R7BP models with and without distance restraints were aligned using Cα carbon alignment in PyMOL. The all-atom RMSD values (no outlier rejection) are shown. **f** Alignment of the bound (blue) and unbound (magenta) models of R7BP without refinement (no outlier rejection) or with refinement show major differences in the head region of the lobster-like structure comprising of both N- and C-termini of R7BP

To further test the model accuracy, we applied XL-MS analysis to the three-dimensional structure of R7BP bound to RGS7/Gβ5 after optimizing DSSO cross-linking conditions (Supplementary Fig. 3). XL-MS analysis of the purified R7BP/RGS7/Gβ5 triplex identified six intramolecular cross-linked sites in R7BP that were identical to a subset of the 14 linked sites identified above from analysis of unbound R7BP (Fig. 1d, Supplementary Table 3). Iterative modeling of R7BP with I-TASSER upon addition of distance restraints to the six cross-linked lysine pairs generated a model of R7BP in the bound state (Fig. 1d). Alignment of Cα carbons with and without distance restraints generated RMSD values of 19.3 and 12.5 for the unbound and bound models, respectively, highlighting a difference in structure upon addition of distance restraints in the I-TASSER modeling (Fig. 1e). The bound and unbound R7BP models were similarly aligned using PyMOL without refinement (no outlier rejections—left) and with refinement (right) and generated RMSD values of 16.7 and 6.7 Å, respectively (Fig. 1f). Distinct differences between the modeled structures with and without restraints is seen, with the most apparent changes observed in the relative positions of the [218]ETM and the [37]GSG regions, which in the bound state are positioned further away from each other. These results suggest that R7BP might exist in two different conformations under bound and unbound conditions.

**R7BP forms a binding groove at R7-RGS/Gβ5 duplex interface.** We next modeled the interaction between R7BP and the R7-RGS/Gβ5 duplex using ClusPro and CHARMM. The reliability of these programs was first assessed by modeling the interactions between human R7-RGS and Gβ5 proteins, as the crystal structures of the mouse RGS9/Gβ5 duplex has been previously resolved[37]. As described above, each member of R7-RGS forms a duplex with Gβ5, similar to the duplex structure of mouse RGS9/Gβ5 (Supplementary Fig. 4a)[37]. To check the reliability of our duplex modeling, we performed Cα carbon alignment of RGS7/Gβ5 and RGS9/Gβ5 models with the available mouse crystal structures and obtained all-atom RMSD values (with no outlier rejection) of 17.1 and 22.3, respectively (Supplementary Fig. 4b). We then modeled the ternary complexes of R7BP with each of the four members of R7-RGS and Gβ5 duplexes using the same strategy. As shown in Fig. 2a–d, in all four triplex models, R7BP appears to form a binding groove along its body from head to tail, primarily at the surface formed by the four main helices (H1–H4), including multiple contact sites with both the DEP and DHEX domains of the R7-RGS proteins[39], as well as the coiled-coil region of Gβ5 (Fig. 2e, Supplementary Fig. 5a).

We next utilized surface plasmon resonance (SPR) biosensor analysis to understand and quantify the interactions between these proteins. We compared the binding affinities of R7BP for the various R7-RGS/Gβ5 duplexes using full-length purified R7BP (R7BP-FL) as the ligand and the four purified R7-RGS/Gβ5 duplexes as separate analytes. The results showed that R7BP-FL had a 10-fold lower binding affinity for the RGS7 duplex than for

**Table 1 SPR analysis of the binding affinities of R7BP variants with R7-RGS/Gβ5 duplexes**

| Ligand | Analyte | $K_D$ (nM) |
|---|---|---|
| R7BP-FL | RGS6/Gβ5 | 2.3 ± 0.8 |
| R7BP-FL | RGS7/Gβ5 | 19 ± 2.6 |
| R7BP-FL | RGS9/Gβ5 | 1.5 ± 0.1 |
| R7BP-FL | RGS11/Gβ5 | 2.3 ± 0.1 |
| R7BPΔN | RGS7/Gβ5 | 1.8 ± 0.6 |
| R7BPΔC | RGS7/Gβ5 | 9.9 ± 0.6 |
| R7BPΔNC | RGS6/Gβ5 | 1.5 ± 0.4 |
| R7BPΔNC | RGS7/Gβ5 | 2.3 ± 0.5 |
| R7BPΔNC | RGS9/Gβ5 | 3.2 ± 0.1 |
| R7BPΔNC | RGS11/Gβ5 | 1.5 ± 0.2 |

the RGS6, RGS9, and RGS11 duplexes (Fig. 2f, Supplementary Fig. 5b, Table 1). This result is in line with previous reports that R7BP showed differential binding affinities to different R7-RGS/Gβ5 duplexes in SPR and immunoprecipitation assays[2,40]. To determine whether the difference in binding affinities was due to buried surface area, the accessible surface area and buried surface area for each component of the R7BP/R7-RGS/Gβ5 triplex was calculated (Supplementary Table 4). No correlation between buried surface area and binding affinity was observed.

To illustrate the structural basis of the lower binding affinity between the RGS7/Gβ5 complex and R7BP, we used RGS7 and RGS9 as representative R7-RGS members and conducted XL-MS analyses of the purified R7BP-FL/RGS7/Gβ5 and R7BP-FL/RGS9/Gβ5 triplexes. We identified four intermolecular cross-linked lysine pairs between R7BP-FL and RGS7/Gβ5 (Fig. 2g) but only one pair between R7BP-FL and RGS9/Gβ5 (Fig. 2h). Interestingly, there were more intramolecular cross-linked sites within Gβ5 in the R7BP-FL/RGS7/Gβ5 triplex (four) than in the R7BP-FL/RGS9/Gβ5 triplex (one) (Supplementary Data 1, Supplementary Tables 5 and 6). These findings suggest that the RGS7 triplex, including Gβ5, adopts a different conformation than that of the RGS9 triplex, which likely contributes to its distinct binding affinity for R7BP-FL.

**R7-RGS structural determinants for R7BP-binding affinity.** The markedly lower binding affinity of the RGS7/Gβ5 duplex for R7BP-FL suggests that RGS7 might contain unique structural features that are responsible for the difference, since the R7BP and Gβ5 species are identical in the four triplexes. Comparison of the amino acid sequences of the four R7-RGS members reveals that RGS7 contains an additional 21 amino acids from aa230 to 250 (RGS7-LP21) that are absent in the corresponding region of RGS9 and RGS11 (Fig. 3a and Supplementary Figs. 1, 6–8). In our models, these extra amino acids form an unstructured loop within the DHEX domain of RGS7 (Fig. 3b, c). The RGS7-LP21 region also engages the N terminus of R7BP in our model and forms a horseshoe-like structure wedged between the R7BP and RGS7

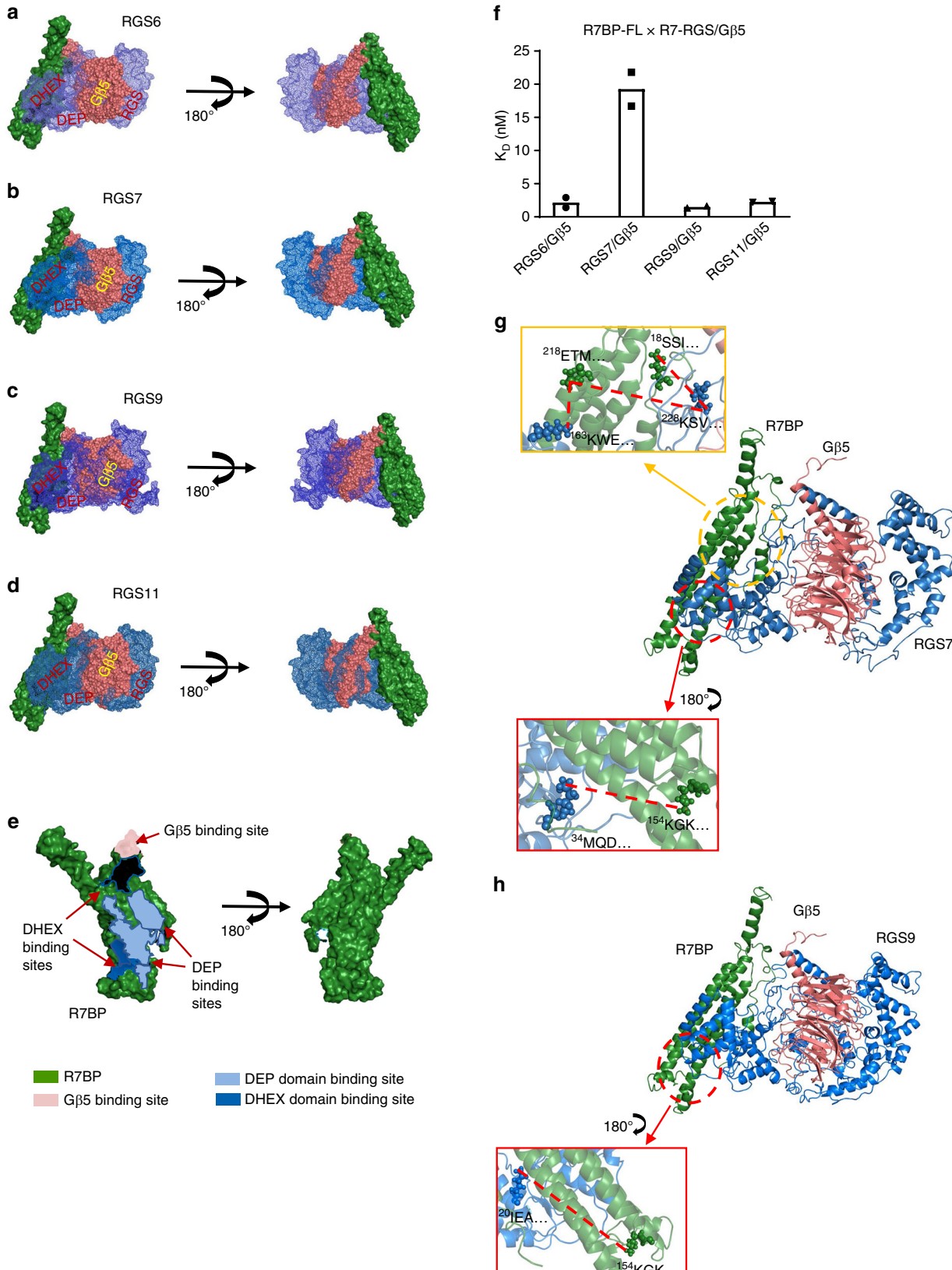

**Fig. 2** XL-MS-calibrated molecular modeling of the R7BP/R7-RGS/Gβ5 triplex. **a–d** Triplex models of R7BP (green surface), Gβ5 (pink spheres), and R7-RGS (blue mesh) involving the R7-RGS proteins obtained through I-TASSER, ClusPro, and CHARMM modeling. **e** R7BP interfaces showing the contact sites for the RGS7 DEP (light blue) and DHEX (dark blue) domains, as well as a potential Gβ5-binding site (pink), as shown in different colors and indicated by arrows. **f** Quantitative analysis of SPR data showing $K_D$ values for interactions of R7BP-FL with each of the R7-RGS/Gβ5 duplexes. Each experiment was repeated twice. **g**, **h** Cartoon models and insets depict the cross-linked inter-molecular lysines and interactions identified between R7BP (green) and RGS7 (G)/RGS9 (H) (blue)

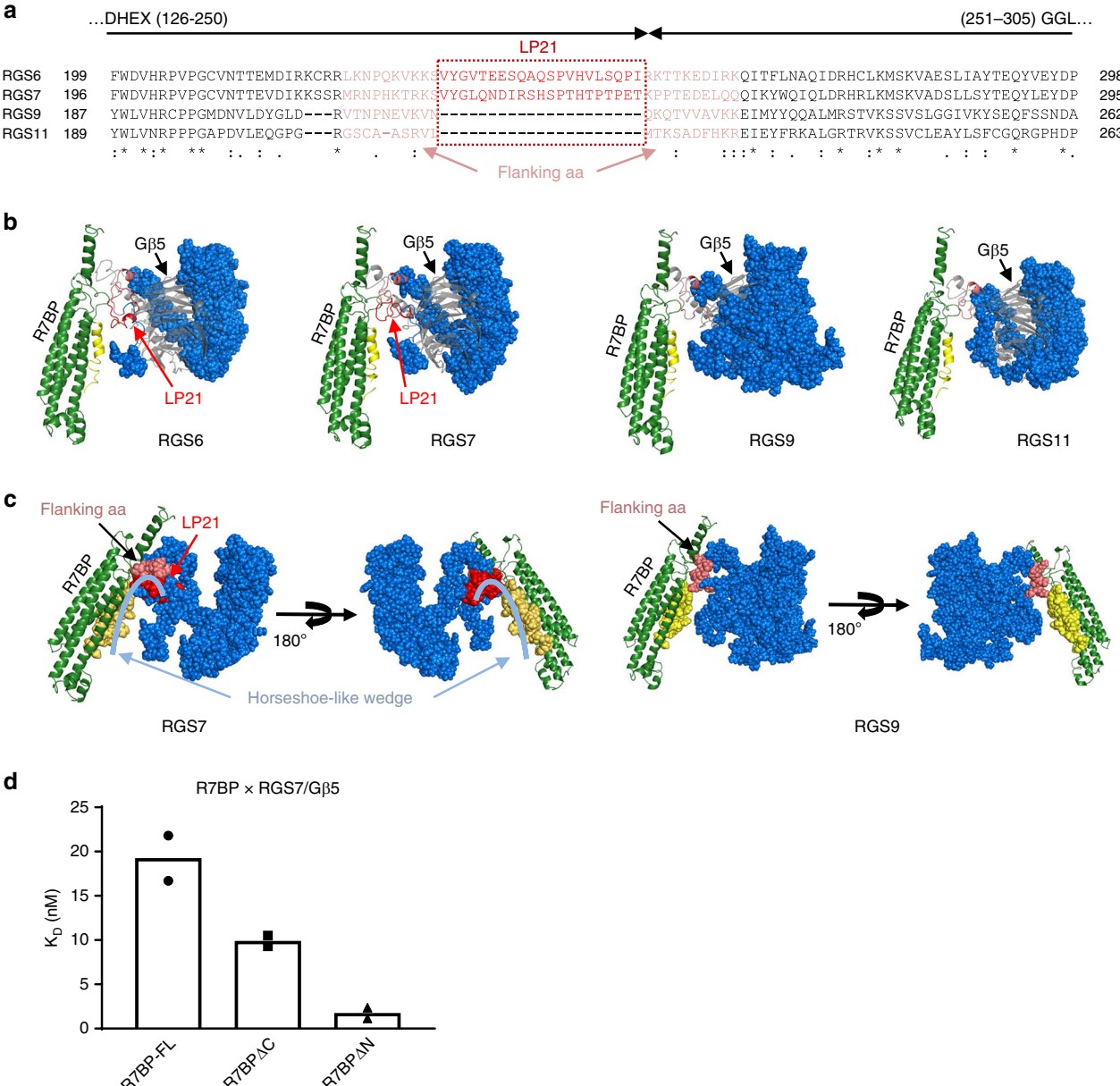

**Fig. 3** Determinants of binding affinity between R7BP and R7-RGS/Gβ5 duplexes. **a** Sequence alignment of residues in the R7-RGS DHEX and GGL domains revealed 21 amino acids that are present in RGS6 and RGS7 (specified in a red frame) but absent in RGS9 and 11. Ten residues flanking either side of the 21-aa region are colored in pink. **b** The modeled 21-aa region (in red) of RGS6 and RGS7 showing a flexible loop (LP21, in red), as compared to the flanking residues (in pink) found in all members of R7-RGS. Only the GGL, RGS, and partial DHEX domains of R7-RGS are shown (blue spheres) for clarity. Gβ5 is shown in gray cartoon and R7BP is shown in green, with the N-terminal 21 amino acids (N21) highlighted in yellow. **c** A horseshoe-like wedge formed between the RGS7-LP21 and the R7BP-N21 is shown in light blue, as compared with RGS9 with only the flanking residues (in pink). **d** Quantification of SPR-binding affinities (in $K_D$ values) of R7BP variants missing either the N- (R7BPΔN) or the C- (R7BPΔC) terminus with RGS7/Gβ5 duplex. Each experiment was repeated twice

interface (Fig. 3b, c). In contrast, in the RGS9 and RGS11 modeled triplexes, the R7BP N-terminus stays unengaged due to the absence of RGS7-LP21 (Fig. 3c). It is therefore tempting to hypothesize that this RGS7-LP21 region is important for the relatively lower binding affinity between R7BP-FL and the RGS7/Gβ5 duplex observed in the SPR analysis. However, in this scenario, the higher binding affinity of the RGS6/Gβ5 duplex for R7BP-FL compared to RGS7/Gβ5 must result from different factors because relative to RGS9 and RGS11, RGS6 has a 21-aa insert (RGS6-LP21) corresponding to RGS7-LP21 (Fig. 3a). We further analyzed the amino acid sequences within these 21-aa

regions and found that RGS7 and RGS6 shared lower-than-average similarities and displayed distinctive electrostatic surface potentials within the region corresponding to LP21 (Supplementary Fig. 9), which could potentially be the cause of their variations in R7BP-FL-binding activities. Another reason for differences in binding affinities may be due to the differences in hydrophobicity of RGS6-LP21 relative to RGS7-LP21 (Supplementary Fig. 10).

Recognition of the putative horseshoe-like structure formed between RGS7-LP21 and the N-terminus of R7BP prompted us to further examine if the R7BP N-terminus also plays a role in the

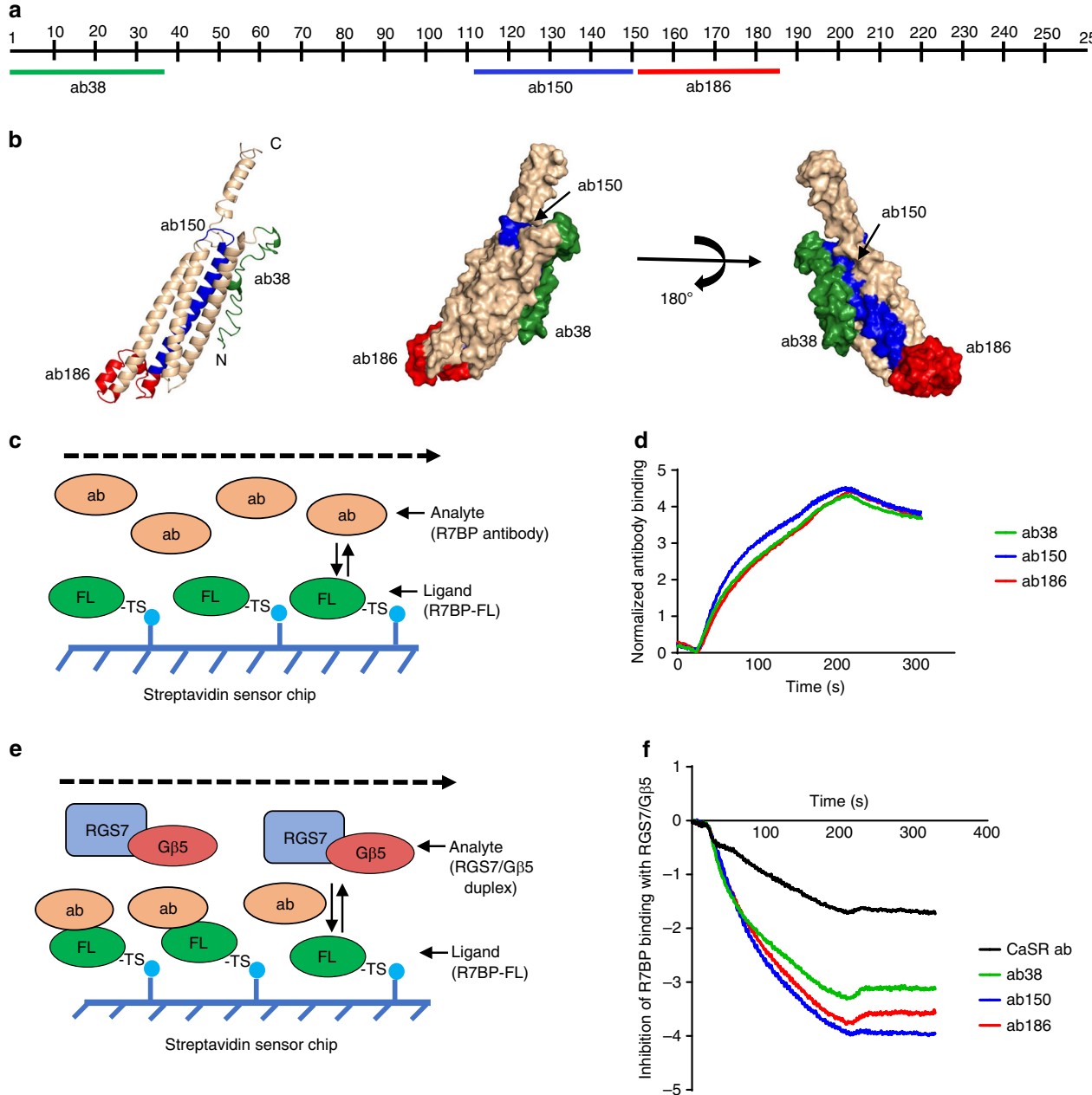

**Fig. 4** Development of antibody blockers disrupting R7BP interactions with RGS7/Gβ5 duplex. **a** Diagram showing the location and sequence information of the R7BP epitopes for the three "active" antibodies ab38 (aa1–38; green), ab150 (aa111–150; blue), and ab186 (aa151–186; red). **b** Mapping of the three epitopes on the R7BP 3D structure model in both cartoon and surface modes, with epitope coloring as in part **a**. **c**, **d** Schematic (**c**) showing SPR analysis of R7BP antibody activities using R7BP-FL as ligand and antibody as analytes and (**d**) normalized binding affinities of each antibody over time. The binding signal for each time point was normalized to the negative control (CaSR ab) signal to obtain the binding curves shown. Source data for part **d** is available in Supplementary Data 2. **e**, **f** Schematic (**e**) showing SPR analysis of antibody inhibition of R7BP-FL binding with the RGS7/Gβ5 duplex using antibodies as the first analyte and the RGS7/Gβ5 duplex as second analyte and (**f**) the inhibition SPR curves of the antibodies over time. CaSR ab was used as control. The binding signals of the RGS7/Gβ5 duplex in the presence of different antibodies at each time point were normalized to the binding signals from the duplex alone to obtain the curves shown. Source data for part **f** is available in Supplementary Data 3

binding affinity of R7BP to the RGS7/Gβ5 duplex. We therefore created a truncated R7BP derivative (R7BPΔN) lacking the N-terminal 21 amino acids. Surprisingly, SPR analysis showed that R7BPΔN had a 10-fold higher affinity for the RGS7/Gβ5 duplex compared to R7BP-FL, resulting in an affinity similar to that between R7BP-FL and the RGS6, 9, or 11/Gβ5 duplexes. Removing the C-terminus of R7BP (aa223–257, R7BPΔC) had much less effect on its binding affinity to RGS7/Gβ5 (Fig. 3d, Supplementary Fig. 11, Table 1). These results strongly suggest

that the N-terminal 21 amino acids of R7BP (R7BP-N21), which form a structure wedged against the RGS7-LP21 region in our model, account for some (or most) of the lowered binding affinity of R7BP-FL vis-à-vis the RGS7/Gβ5 duplex. It is intriguing to postulate that the horseshoe-like structure formed between R7BP-N21 and RGS7-LP21 might create some extra space at the interface, resulting in reduced binding affinity and increased accessibility to the DSSO cross-linking reagent. The different binding affinities with R7BP-FL between RGS6/Gβ5 and

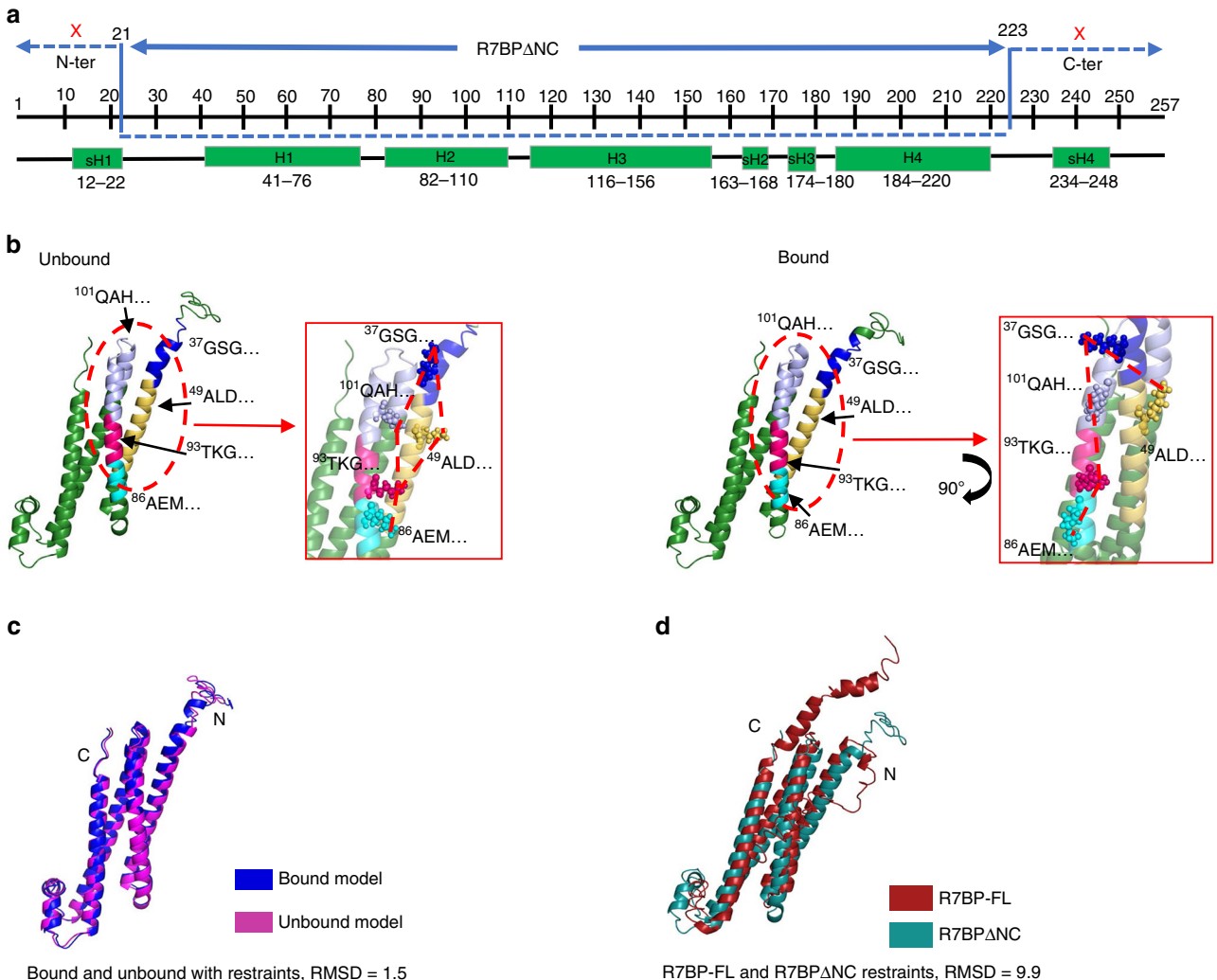

**Fig. 5** XL-MS-calibrated molecular modeling of R7BPΔNC. **a** Diagram showing R7BPΔNC location and the relative positions of the deleted sequences at both N and C termini of R7BP. **b** I-TASSER modeling of the R7BP-ΔNC structure in the unbound state and when bound to RGS7/Gβ5. The cross-linked lysines and interactions are highlighted in the insets. **c** The bound and unbound models were aligned using Cα carbons to obtain an RMSD of 1.5. **d** Overlay of the R7BP-FL (maroon) and R7BPΔNC (teal) models show the greatest differences in their N- and C-terminal unstructured regions. The Cα carbon RMSD alignment score is 9.9

RGS7/Gβ5 duplexes indicated that it is potentially possible to amend their binding affinities through manipulating the amino acid sequences within and/or adjacent to either RGS6-LP21 or RGS7-LP21.

**Development of antibody inhibitors of R7BP.** In light of the critical role of R7BP-N21 in determining the binding affinities between R7BP-FL and the RGS7/Gβ5 duplex through interactions with RGS7-LP21 demonstrated in the experiments above, we explored the possibility of blocking R7BP function and R7-RGS/Gβ5 duplex interaction through antibodies targeting the R7BP N-terminal region. We first generated polyclonal llama antibodies against the N-terminal 38 amino acids of R7BP (ab38) and using R7BP-FL as ligand and ab38 as analyte, we observed a high-affinity interaction by SPR (Fig. 4a–d). We then tested the antibody's ability to inhibit the interactions between R7BP-FL and the RGS7/Gβ5 duplex by SPR, also using R7BP-FL as the ligand. The epitope sites of R7BP were first saturated with excess ab38 or control antibody (CaSR-ab: llama polyclonal antibody generated against the extracellular domain of the calcium sensing receptor) as the first analyte prior to applying the RGS7/Gβ5 duplex as the

second analyte. As shown in Fig. 4e, f, ab38 dramatically blocked R7BP interactions with the RGS7/Gβ5 duplex relative to control antibody, indicating the potential to be an effective inhibitor of R7-RGS/Gβ5-dependent R7BP function.

Encouraged by the inhibitory effects of ab38, we explored whether we could inhibit other interaction sites of R7BP-FL and the RGS7/Gβ5 duplex using llama antibodies targeting different regions of R7BP (Fig. 4a, b). Indeed, the antibodies against aa111–aa150 (ab150) and aa151–aa186 (ab186) displayed high binding affinities for the R7BP-FL protein and were also able to strongly block R7BP interactions with the RGS7/Gβ5 duplex (Fig. 4c–f). In contrast, antibodies raised against other regions of R7BP (ab74, ab110, ab222, and ab257) displayed lower affinities with R7BP-FL and were therefore not tested for their inhibitory effects (Supplementary Fig. 12). Antibodies were quantified relative to the R7BP/RGS7/Gβ5 triplex and they specifically interacted with only R7BP in the presence of all three proteins (Supplementary Fig. 13). Because the epitopes of ab150 and ab186 span the H3–sH2–sH3–H4 region of R7BP, which forms the tail of the homarine R7BP structure (Fig. 1a, b), their dramatic inhibitory effects on the interactions of R7BP-FL with

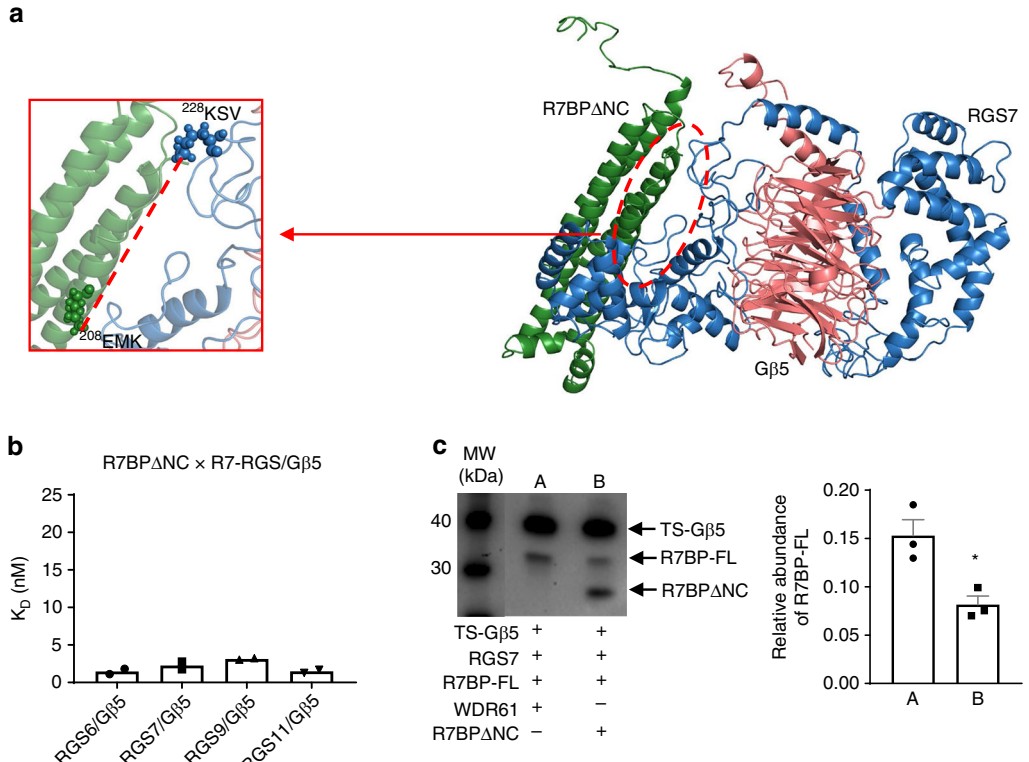

**Fig. 6 Dominant negative inhibition of R7BP interactions with RGS7/Gβ5 duplex. a** Modeling and XL-MS analysis of the R7BPΔNC (green)/RGS7 (blue)/Gβ5 (pink) triplex shows one cross-linked peptide between RGS7 and R7BPΔNC. **b** Quantification of the binding affinities (in $K_D$ values) of R7BPΔNC with each of the R7-RGS/Gβ5 duplexes using SPR. Each experiment was repeated twice and analyzed using Student's t-test. **c** Experiment demonstrating the inhibitory role of R7BPΔNC, as the dominant negative diminishes the binding of R7BP-FL to the RGS7/TS-Gβ5 duplex in Expi293 cells. The expression of different proteins was shown by Western blot and the amount of R7BP-FL pulled down by the RGS7/TS-Gβ5 duplex was quantified. Each experiment was repeated three times and analyzed using Student's t-test (*$p < 0.05$). The source data for part **c** is available in Supplementary Data 4

the RGS7/Gβ5 duplex is consistent with the hypothesis stated above that this region could be the primary docking site for the RGS7 DEP domain, as shown in Fig. 2e.

**Dominant negative as potential inhibitors of R7BP.** Above we have shown that deletion of the R7BP-N21 region enhanced R7BP-binding affinity toward the RGS7/Gβ5 duplex, while deletion of its C terminus had an additional, but more limited, enhancing effect. The C terminus of R7BP containing two palmitoylation sites is essential for its membrane attachment and facilitates the GAP activities of the RGS7/Gβ5 duplex[4]. These observations inspired us to propose that the truncated central part of R7BP, missing both N- and C-termini, R7BPΔNC (Fig. 5a), would have an increased binding affinity for the RGS7/Gβ5 duplex, similar to or higher than R7BPΔN, and could thus serve as a potential dominant negative inhibitor of the interactions of R7BP with the RGS7/Gβ5 duplex.

Similar to R7BP-FL, we performed XL-MS on R7BPΔNC and applied distance restraints when performing I-TASSER modeling to obtain the unbound structure shown in Fig. 5b (left) and Supplementary Table 7 (see also Supplementary Data 1). The R7BPΔNC/RGS7/Gβ5 triplex was also subject to XL-MS followed by modeling of R7BPΔNC to obtain the bound structure shown on the right in Fig. 5b and Supplementary Table 8 (see also Supplementary Data 1). Alignment of bound and unbound models (Fig. 5c, Supplementary Fig. 14a) indicated no major difference in structure and orientation of the proteins, implying that the N and C terminal regions lend the most flexibility to the

R7BP structure. This is highlighted by alignment of R7BP-FL and R7BPΔNC models, which revealed that conformational changes appear mainly in the "head" of the lobster-like structure, i.e., the region in R7BP-FL predicted to interact with the DHEX region of R7-RGS (Fig. 5d).

XL-MS analysis of the R7BPΔNC/RGS7/Gβ5 triplex showed much tighter intramolecular interactions, similar to those in the R7BP-FL/RGS9/Gβ5 triplex, and fewer DSSO accessible sites, as shown in Fig. 6a and Supplementary Table 9. Modeling of the R7BPΔNC/RGS7/Gβ5 triplex revealed that the interface between R7BPΔNC and the RGS7/Gβ5 duplex was much like that with R7BP-FL. As predicted, SPR analysis indicated that R7BPΔNC has approximately 10-fold higher affinity to the RGS7/Gβ5 duplex compared to R7BP-FL, similar to the RGS7/Gβ5 affinity of the R7BPΔN variant, whereas its binding affinity toward the RGS6, 9, and 11/Gβ5 duplexes remained mostly unchanged (Fig. 6b, Supplementary Fig. 14b, Table 1). The SPR data further reinforced the view that R7BP-N21 is a critical determining factor for R7BP-binding affinity with the RGS7/Gβ5 duplex, but not for the other duplexes. As shown in Fig. 3a, sequence alignment suggests there is a "missing" region in RGS9 that corresponds to RGS7-LP21, a part of the DHEX domain that interacts with R7BP-N21. It is tempting to postulate that these interactions determine the R7BP binding differences for RGS7 and RGS9 when duplexed with Gβ5, since the absence of R7BP-N21 increases its RGS7/Gβ5-binding affinity to a level comparable to that of RGS9/Gβ5. Therefore, interactions involving the N-terminal region of R7BP might account for the previously reported selectivity of

R7BP between RGS7 and RGS9 in instances where they are co-expressed[2].

The markedly enhanced binding affinity of R7BPΔNC with the RGS7-Gβ5 duplex suggested a unique opportunity to specifically disrupt RGS7/Gβ5 duplex functions using R7BPΔNC as dominant negative because it would be less effective to compete with other R7-RGS/Gβ5 duplexes with similar binding affinities. To explore this possibility in vivo, we co-expressed R7BPΔNC and R7BP-FL together with RGS7 and TS-Gβ5 and used TS-Gβ5 to pull down the triplex. As shown in Fig. 6c (and Supplementary Fig. 14c), in the presence of R7BPΔNC, the binding of R7BP-FL with the RGS7/Gβ5 duplex was reduced ~50% ($p < 0.02$), indicating that R7BPΔNC could potentially be used as a dominant negative to effectively block R7BP-FL functions.

## Discussion

Determination of high-resolution structure of proteins and multimeric complexes, e.g. by X-ray spectroscopy or cryo-EM, can shed light on some of the critical interfaces that may play a role in a given PPI network. These three-dimensional structures can be used to gain atomic-level insight into the nature of the interactions and guide the development of modulators, including inhibitors, at one or more critical steps within the corresponding networks[19,20,41]. Finding suitable conditions for protein crystallization is challenging, which hinders progress towards the physicochemical characterization of PPI interfaces and impedes the rational design of drugs. Still, the interfaces observed in a crystal structure are only part of a rich spectrum of possibilities in which proteins interact with one another in an aqueous solution. Recent advances in NMR-based techniques, for example, have shown that proteins interact transiently at multiple sites through ultra-weak, non-specific interactions[42]. These associations are difficult to detect experimentally and are unlikely to be captured in a crystal but are thought to play a role in molecular recognition. Therefore, they can also be legitimate drug targets within a PPI network. In general (see refs. [43,44], and references therein), proteins interact through strong, weak, and ultraweak forces. The entire range of such forces plays both morphological and functional roles, helps mediate specific and non-specific associations, and governs interactions (often multifunctional) through multiple binding interfaces and with different partners throughout the cell cycle.

Ab initio prediction of protein structure (i.e., from its primary sequence) and of multiprotein complexes (i.e., from the three-dimensional structures of the individual proteins) have long been active areas of research in computational biophysics. Although challenging due to limitations in the quality of the forcefield and sampling technique, as well as in computer technology, conceptual and practical progress are being made on all these fronts [43–45]. In this study, however, we have substantially simplified the general problem involving the hard-to-crystalize[46,47] R7BP protein by resorting to partial structural information obtained experimentally. The experimental data was then incorporated into the modeling process through an iterative refining protocol that makes use only of publicly available software. This approach is in line with recent advancements in integrative structural biology that incorporate molecular modeling with XL-MS and provide a viable and simpler alternate to gain valuable insight into protein structures and PPI interfaces[31,48] in the absence of high-resolution structures. The method, relatively straightforward and easy to implement, led us to propose a "lobster-like" 3D structure of the bound and the unbound R7BP. The resulting model has an elongated surface interface that interacts with both the DEP and

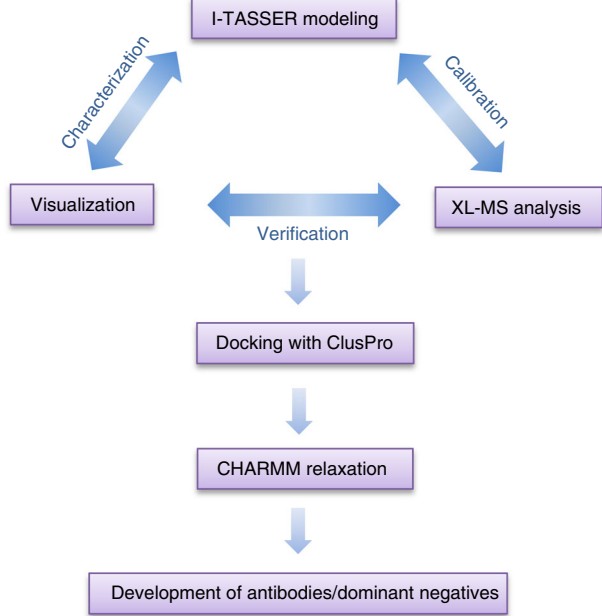

**Fig. 7** Diagram of the XL-MS-calibrated molecular modeling method. Summary of the XL-MS/modeling hybrid approach used in this study. Proteins were initially modeled with the I-TASSER server, and subsequently refined through iterative adjustment that incorporated distance restraints between cross-linked peptides obtained from XL-MS experiments. This process was repeated until the calibrated models were consistent with XL-MS data. To visualize two or more protein interactions, the docking program ClusPro was used to obtain models of the protein complexes that agreed with XL-MS distance restraints. These protein models were structurally relaxed through CHARMM energy minimization yielding the final structures. The complexes were subject to the ultimate experimental test by successfully designing sequence-specific antibodies targeting the proposed binding sites

DHEX domains of R7-RGS/Gβ5 duplexes, consistent with previous predictions based on in vitro mutagenesis and biochemical experiments[39,49]. Through identifying the essential interaction regions of R7BP-N21 and RGS7-LP21, as well as the tail region of the "lobster-like" structure, we successfully developed first-generation R7BP antibodies to the corresponding interaction regions that effectively obstruct its binding to the RGS7/Gβ5 duplex by more than 50%. Similar blockade of R7BP-RGS7/Gβ5 interactions was also demonstrated using a dominant negative structure, R7BPΔNC, in vivo as expected. These results demonstrate that the XL-MS-calibrated molecular modeling approach, which employs few, judiciously chosen, readily available software, is an effective way to obtain relevant structural information for the development of PPI inhibitors.

Many of the structural inferences generated from our experiments utilizing this comprehensive approach, summarized in Fig. 7, were further confirmed by the newly resolved RGS7/Gβ5 duplex crystal structure and its potential interaction models with R7BP obtained from hydrogen–deuterium exchange (HDX) data[35]. It is worth mentioning that our approach generated structural information in the RGS7-LP21 region (RGS7 aa230–250), yet the corresponding region in the crystal structure of the RGS7/Gβ5 duplex (RGS7 aa219–251) was missing[35]. The stepwise iterative modeling employed in this study might have broad applications, as it may help understand the structural basis of other systems for which crystallographic data of key proteins is incomplete or unavailable, thereby

accelerating the drug development process targeting clinically relevant PPIs.

Due to the limitations of small molecules in modulating PPIs in protein complexes, macromolecules, including interfering peptides and macrocyclic peptides[50], that can act as PPI modulators for protein partners with large or shallow interfaces, are attractive alternatives to next-generation drugs[16,51,52]. The advancements in peptide administration and modification technologies[16] and the recent FDA approval of AAV vectors as delivery tools in gene therapy further inspired interest in peptide inhibitors, especially for intracellular PPI targets. In this study, we demonstrated the utility of our comprehensive approach through the development of inhibitory antibodies and a dominant negative against R7BP. This approach diminishes the need for protein crystal structures and bypasses the time-consuming and costly screening of candidate peptides through phage display or other expression libraries[17,51,52], and likely represents one of the most economical and efficient ways to develop interfering peptide inhibitors of PPIs. We further showed that SPR is a reliable method to probe and test the PPIs under study and quantify the binding affinities of various protein partners. Such an approach can shed light on potential interaction mechanisms and reveal selectivity among different partners as demonstrated here between full-length and mutant R7BP and duplexes containing the different R7-RGS proteins. Both first-generation polyclonal antibodies and dominant negative constructs can be used as stepping stones for future development of more effective and specific antibodies. For example, in itch treatment, minimized protein domains, such as single domain antibodies or interfering peptides mimicking antibody-binding sites could be used as effective inhibitors of critical R7BP functions. Potentially, the DNA coding sequence derived from interfering peptides, humanized single domain antibody, or an R7BP-derived dominant negative construct could be cloned into AAV vectors for intracellular expression and inhibition of its interactions with the R7-RGS/Gβ5 duplex, thus providing a novel alternative for the effective and precise treatment of itch with minimum side effects.

## Materials and methods

**DNA constructs**. The human genes encoding *R7BP*, *GNB5* (short form), *RGS6*, *7*, *9–2*, and *11* were used in this study. For all *R7BP* constructs, the palmitoylation site for membrane association was removed by site-directed mutagenesis (C252S, C253S) to facilitate expression and purification (QuikChange II XL site-directed mutagenesis kit, Agilent). Other modifications of *R7BP* constructs include the C-terminal additions of either the TEV protease site and Twin-Strep tag (IBA LifeSciences) to create the *R7BP-TEV-TS* construct or 6x HIS tag to create the *R7BP-HIS* construct. *R7BP* was also N-terminally tagged with either TS or 6x HIS to create *TS-R7BP* and *HIS-R7BP*, respectively. The truncated version of *R7BP*, *R7BPΔNC-TEV-TS*, was created by removing the N-terminal 20 amino acids and C-terminal 34 amino acids. All DNA constructs were cloned in pcDNA3.1 vector in the lab, except *HIS-R7BP*, which was in pcDNA3.1 and was purchased from Bio Basic Inc., and both *RGS7* and *RGS9-2* in pCMV3 vector from Sino Biological Inc. For *GNB5* constructs, the *GNB5* DNA was purchased from cDNA Resource Center and cloned into pcDNA3.1 with the N-terminal addition of either 6x HIS or TS tag to generate *HIS-GNB5* and *TS-GNB5*, respectively.

**Cell culture**. Expi293 HEK cells (purchased new from ThermoFisher Scientific) were used for all gene expression experiments following the manufacturer's instructions. Briefly, cells were grown in Expi293 expression medium (Thermo Fisher Scientific) in 250 mL vented round flasks with shaking at 125 rpm in a 37 °C/ 8% CO$_2$ incubator. DNA transfections were conducted in overnight cultures at a cell density of ~2.5 × 10$^6$ cells per mL using the ExpiFectamine 293 transfection kit (Thermo Fisher Scientific). After addition of transfection enhancing reagents 16–18 h post transfection, cells were grown for another 48 h before collection by centrifugation at 500 rpm for 5 min. The cell pellets were stored at −80 °C until use.

**Protein purification**. Frozen cell pellets were thawed and resuspended in either His lysis buffer (50 mM Tris–HCl, pH 7.4, 150 mM KCl, 10 mM imidazole, 7.5% glycerol) or Strep lysis buffer (50 mM Tris, pH 7.4, 150 mM KCl, 7.5% glycerol) supplemented with 0.5 mM TCEP, protease inhibitor tablets and PhosSTOP tablets (Roche, 2 tablets per 10 mL lysis buffer) at a ratio of 5 mL lysis buffer per 30 mL cell culture pellet. The lysis of cells was performed using a homogenizer (Omni Tissue Homogenizer) for 30 s every 6 min for a total of six pulses. The cell debris was removed by low speed centrifugation at 7500 × *g* for 30 min at 4 °C, followed by high speed centrifugation of the supernatant at 24,000 × *g* for 30 min at 4 °C. After passing through a 0.45-micron filter, the supernatant was applied to either a 1 mL column of HisTrap HP (GE Healthcare LifeSciences) for HIS-tagged proteins or StrepTrap HP (GE Healthcare LifeSciences) for TS-tagged proteins at a rate of 0.3 mL per minute using an AKTA protein purification system (GE Healthcare LifeSciences). The column was washed with 10 mL wash buffer for either HIS-tagged proteins (100 mM Tris–HCl, pH 8.0, 150 mM KCl, 20 mM imidazole, 0.5 mM TCEP) or TS-tagged proteins (100 mM Tris, pH 8.0, 150 mM KCl, 0.5 mM TCEP), followed by 20 mL wash buffer supplemented with 10 mM MgCl$_2$ and 10 mM ATP. After a final wash with 20 mL of wash buffer containing 4.5 M NaCl, the tagged proteins were eluted from either the HIS column with imidazole elution buffer (100 mM Tris–HCl, pH 8.0, 150 mM KCl, 500 mM imidazole, 0.5 mM TCEP), or the Strep column using the desthiobiotin elution butter (100 mM Tris, pH 8, 150 mM KCl, 0.5 mM TCEP, 20 mM desthiobiotin (IBA Lifesciences)). If needed, a size-exclusion chromatography high-resolution column (HiPrep 16/60 Sephacryl S-300, GE Healthcare LifeSciences) was used to further purify the proteins. All proteins were buffer exchanged into storage buffer (20 mM Tris, pH 7.4, 150 mM KCl, 0.5 mM TCEP, 5% glycerol) and stored at −80 °C until further use. The purity of the eluted proteins was examined by SDS–PAGE analysis.

**Antibody purification**. Llama polyclonal R7BP antibodies were generated using purified R7BP-TEV-TS protein by Kent Laboratories. The rationale for using llama serum rather than other sources for antibody production and isolation was the large yield and time and cost effectiveness of this procedure. Seven peptides (35–45 amino acids long) spanning the entire sequence of R7BP were synthesized, each with a Twin-Strep tag (SAWSHPQFEK(GGGS)$_2$GGSAWSHPQFEK), from either GeneScript or Peptideamerica and used for isolation of each corresponding antibody from llama serum as described below: 10 mL of serum was incubated with 100 µg of peptide at 4 °C overnight with slow rotation and applied to a ~200 µL Strep-Tactin Sepharose column (IBA Lifesciences). The column was washed with 5 mL of Strep Wash buffer containing 4.5 M NaCl, followed by 2 mL of Strep Wash buffer only. The antibody was eluted with 500 mL of Strep Elution buffer containing 30 mM desthiobiotin (pH 8). The eluted antibodies were separated from the peptides by applying them to a spin column and washed with 2 × 500 µL of 100 mM glycine buffer (pH 2.7, HCl) and neutralized with 3 × 500 µL storage buffer. The negative control llama antibody used for SPR, CaSR (Calcium Sensing Receptor) antibody, was similarly obtained using the purified extracellular domain of CaSR-HIS protein. All purified antibodies were quantified by SDS–PAGE gel electrophoresis and stored in storage buffer at 4 °C until use. Alternatively, in order to remove excessive albumin from the serum and obtain higher antibody yields, total IgGs were isolated from llama serum using the caprylic acid purification method[53] by adjusting the serum pH to 5.5 and stirring with caprylic acid for 90 min, followed by centrifugation. The purified IgG was used for peptide-specific antibody purification as described above. Antibodies were quantified by loading 1 µg total purified protein on a gel and quantifying the band intensity relative to a known concentration of purified R7BP/RGS7/Gβ5 triplex. To test for antibody specificity, Western blot was performed using purified R7BP (no tag) and RGS7/HIS-Gβ5. Ab38, ab150, and ab186 were incubated with the membrane and a band corresponding to the correct molecular weight was detected in the R7BP lane. No bands were detected for the RGS7/Gβ5 duplex, highlighting that there was no cross-reactivity between the antibodies raised against R7BP and the RGS7/Gβ5 duplex. Planned future experiments will involve fine mapping of antibody epitopes and the development of specific monoclonal antibodies to pinpoint and block interaction sites.

**Cross-linking mass spectrometry (XL-MS)**. Proteins were prepared for XL-MS analysis as detailed in Kao et al.[54]. Briefly, a 20 mM solution of DSSO (Thermo Fisher Scientific) was mixed with purified proteins in PBS (pH 7.4) at ratios of 1:100, 1:200, or 1:300 fold (protein:cross-linker) and incubated at room temperature for one hour. The reaction was quenched by addition of 1 M Tris–HCl, pH 8 to a final concentration of 20 mM. The cross-linked proteins were separated on an SDS–PAGE gel, stained with SimplyBlue SafeStain (Thermo Fisher Scientific) for one hour and de-stained in water overnight. The cross-linked bands at expected size were excised from the gel and further de-stained by addition of 100 µL Wash Solution (25 mM NH$_4$HCO$_3$ in 50% acetonitrile) and incubated at room temperature with slow shaking for 10 min. The de-staining process was repeated several times until the gel pieces became colorless, then the gel pieces were completely dried in a Speed Vac. Reduction solution (10 mM DTT in 25 mM NH$_4$HCO$_3$) was added to the dried gel pieces and incubated at 56 °C for 30–60 min. After removal of the reduction solution, 50 µL of alkylation solution (55 mM iodoacetamide in 25 mM NH$_4$HCO$_3$) was added and incubated in the dark at room temperature for 30–45 min. The gel pieces were washed with ~100 µL 25 mM NH$_4$HCO$_3$ for 2 × 10 min and dried completely in a Speed Vac. In-gel digestion of proteins was performed by incubation with ~20 µL trypsin solution (12.5 ng µL$^{-1}$ trypsin (Promega) in 25 mM ice-cold NH$_4$HCO$_3$) at 4 °C for 30 min. After removing the trypsin, 25 mM NH$_4$HCO$_3$ was added to cover the gel pieces and

incubated at 37 °C overnight. The supernatant was saved and a solution of 50% acetonitrile/5% formic acid (enough to cover) was added to the gel pieces and incubated on a shaker for 20–30 min at room temperature. The supernatant was saved, and this step was repeated once. The saved supernatants were dried in a Speed Vac and resuspended in 20 μL 0.1% formic acid in water. The peptides were desalted using a C18 ZipTip (MilliporeSigma), dried in the Speed Vac, resuspended in 0.1% formic acid and transferred to an LC–MS autosampler vial.

**Liquid chromatography**. Tandem mass spectrometry (LC–MS/MS) was performed using a Dionex UltiMate 3000 rapid separation nano UHPLC system (Thermo Fisher Scientific) coupled online to an Orbitrap Fusion Lumos tribrid mass spectrometer (Thermo Fisher Scientific). Tryptic digests of DSSO crosslinked proteins was first loaded onto a nano trap column (Acclaim PepMap100 C18, 3 μm, 100 Å, 75 μm i.d. × 2 cm, Thermo Fisher Scientific), and then separated on a reversed-phase EASY-Spray analytical column (PepMap RSLC C18, 2 μm, 75 μm i.d. × 50 cm, Thermo Fisher Scientific) using a linear gradient of 4–32% B (buffer A: 0.1% formic acid in water; buffer B: 0.1% formic acid in acetonitrile) for 100 min. The mass spectrometer was equipped with a nano EASY-spray ionization source, and eluted peptides were brought into gas-phase ions by electrospray ionization and analyzed using an MS2-MS3 strategy. High-resolution survey MS scans and CID fragment MS2 spectra were acquired in the orbitrap in a data-dependent manner with a cycle time of 5 s. Dynamic exclusion was enabled. Signature peaks in MS2 indicative of individual peptides originating from a cleaved DSSO-crosslinked peptide pair were further isolated for HCD MS3, and fragment ions were recorded in the ion trap. Raw data files generated from LC–MS/MS were analyzed using a Proteome Discoverer software package (Thermo Fisher Scientific) and the Sequest HT search engine. The following database search criteria were set to: enzyme, trypsin; max miscleavages, 3; variable modifications, DSSO Alkene (K), DSSO Thiol (K), oxidation (M), deamidation (NQ); precursor mass tolerance, 20 ppm; fragment mass tolerance, 0.5 Da. Peptide-spectrum matches (PSMs) were filtered to achieve an estimated false discovery rate (FDR) of 1% based on a target-decoy database search strategy. A custom in-house script was employed to generate a list of crosslinked peptide pairs and their corresponding proteins.

The samples were prepared with purified proteins, which greatly reduced the chance of false positives that were assigned to our target proteins but were in fact derived from contaminants or other unwanted components in a sample. The latter is more likely when working with a complex proteome. During database searches, the whole protein sequence database for human was used, which also considerably lowered the probability of getting false-positive assignments in comparison with some reported approaches using a small database consisting of only a few target protein sequences. In addition, the crosslink pairs were identified with multiple PSMs and in many cases at different charge states. Furthermore, they were also seen consistently in different experiments, which substantially increased our confidence in the crosslink pairs identified. Manual examination of spectra was also applied when appropriate. All proteins were cross-linked with DSSO and processed and analyzed as described above. Each protein was incubated with DSSO (either 1:100, 1:200, or 1:300 protein:DSSO ratios) and we repeated the experiment once at a second ratio. The cross-linked peptide data obtained from both experiments was combined to compile a list of total cross-linked peptides for each protein interaction. Detailed information on all identified cross-links and other information such as $m/z$ ratios, ion score, and charge are available on PeptideAtlas. org (http://www.peptideatlas.org/PASS/PASS01391).

**SPR spectroscopy**. SPR experiments were performed using the OpenSPR instrument and Streptavidin sensor chip from Nicoya Lifesciences. All proteins were buffer exchanged into PBS-P buffer (PBS pH 7.4, 0.005% Tween20). The ligand (tagged protein) was injected into the port and attached to the chip, followed by injection of the analyte (interacting protein). The flow rate was maintained at 25 μL per minute and the buffer was flowed over the chip for an additional 4 min after terminating the flow of analyte. The chip was regenerated with the appropriate buffer and the experiment was repeated at different analyte concentrations. SPR data was analyzed using TraceDrawer software (Nicoya Lifesciences) assuming a 1:1-binding model. All $K_D$ values were calculated from two independent experiments and 3–5 concentrations per analyte. Antibody binding and blocking assays were performed three times for each antibody. R7BP-TEV-TS was attached to the chip as ligand, followed by injection of RGS7/Gβ5 as analyte to determine optimal binding signal. After regeneration with 5 mM NaOH, 20 μg mL$^{-1}$ antibody was injected and the binding signal to R7BP was measured. Next, the same concentration of duplex was injected and the signal measured was compared with that produced prior to antibody injection to obtain the blocking capacity of each antibody against the R7BP-RGS7/Gβ5 interaction. For each experiment, the signal values for each antibody were normalized to the CaSR antibody (negative control) signal, while the RGS7/Gβ5 signal values post-antibody injection were normalized to the RGS7/Gβ5-binding signals to R7BP on the chip prior to antibody injection. The antibody signals for each time point were averaged to generate the time response curves.

**In vivo competition assay**. To determine whether truncated R7BP, R7BPΔNC, that has higher binding affinities to the RGS7/Gβ5 duplex, can be used as a

dominant negative in vivo, Expi293 cells were transfected with a mixture of plasmids encoding RGS7, TS-Gβ5, R7BP-HIS, and either WDR61 (sample A) or R7BPΔNC (sample B), respectively. Pelleted cells were resuspended in lysis buffer (100 mM Tris pH 7.4, 150 mM NaCl, 1 mM EDTA, 7.5% glycerol) and lysed using a BeadBlaster 24 (Benchmark) with BeadBug beads (MilliporeSigma). The lysate was centrifuged at 13,000 × g for 20 min and the supernatants were re-spun for 5 min. The lysates were applied to 2 μL of pre-washed MagStrep "type3" XT Beads (IBA Lifesciences) and incubated with gentle rotation at 4 °C for one hour. Supernatants were collected, and the beads were washed three times with wash buffer (100 mM Tris pH 8.0, 150 mM NaCl, 1 mM EDTA) and eluted in 50 μL buffer BXT (100 mM Tris pH 8.0, 150 mM NaCl, 1 mM EDTA, 50 mM biotin) with gentle shaking for 10 min. The elution step was repeated once to obtain higher yields. The total protein concentration was measured and analyzed by SDS–PAGE using 1 μg of protein for each sample. Western blot was performed by transferring the proteins to a nitrocellulose membrane using the iBlot2 dry gel transfer apparatus (Thermo Fisher Scientific) and probed with anti-Gβ5 (ATDG) antibody (in-house purified, 2 mg mL$^{-1}$) followed by anti-R7BP (SAP-35) antibody (in-house purified, 2 mg mL$^{-1}$). Signal images were captured using the ChemiDoc MP Imagining System (BioRad) and quantified using ImageJ[55].

**Protein modeling**. The RGS9-Gβ5 mouse protein crystal structure (PDB code: 2PBI) and RGS7-Gβ5 mouse crystal structure (6N9G) were used to model all R7-RGS/Gβ5 structures. Each human R7-RGS sequence and the human Gβ5 sequence were input into the I-TASSER online server. I-TASSER retrieves proteins of similar secondary structures and folds from the PDB database in order to model protein structures. Since RGS7/Gβ5 and RGS9/Gβ5 mouse crystal structures already exist in the PDB library, R7-RGS proteins and Gβ5 can be individually modeled even though they exist as an obligate dimer in the cell. The resulting human models were aligned with the above crystal structures using PyMOL to obtain each R7-RGS/Gβ5 preliminary structure. Next, we performed XL-MS on RGS7/Gβ5 and RGS9/Gβ5 and collected information about proximal lysine residues in the duplex. The I-TASSER results were refined by assigning distance restraints of 10 Å (DSSO cross-linker spacer arm length) to the N-Z (N-ζ) atoms between each cross-linked lysine. These models were relaxed using CHARMM to obtain the best fit models for each R7-RGS/Gβ5 duplex. Chemistry at HARvard Macromolecular Mechanics (CHARMM) is a molecular dynamics simulation program that runs a potential energy minimization simulation of a given structure. It resolves steric hindrance and other unfavorable molecular interactions to create a new structure with reduced potential energy. Similarly, the R7BP sequence was input into I-TASSER with lysine distance restraints information, followed by CHARMM, to obtain the R7BP model.

The R7BP/R7-RGS/Gβ5 triplex was modeled by introducing the PDB coordinates (obtained and refined from I-TASSER) for R7-RGS/Gβ5 and R7BP as receptor and ligand, respectively, into the ClusPro online server, with the addition of distance restraints of 10–20 Å for cross-linked lysine residues. The addition of distance restraint information yielded 10 models of the triplex, with R7BP oriented in the expected position relative to R7-RGS (near the DEP/DHEX domain). This is in contrast to ClusPro analysis without distance restraint data, which yielded 30 triplex models with R7BP oriented in various positions relative to the R7-RGS/Gβ5 duplex. Thus, addition of the distance restraint data obtained through XL-MS greatly aided in obtaining more reliable models for the triplex. The best triplex structure out of these 10 models was selected by analyzing the distance information for each cross-linked lysine pair in the triplex and selecting the model with minimal distances between lysines. After performing a short energy minimization with CHARMM, the validity of the final, relaxed models was assessed by comparison with the corresponding XL-MS data. Multiple sequence alignments were performed with Clustal Omega[56].

**Statistics and reproducibility**. SPR experiments to calculate $K_D$ values were performed in duplicate with 3–5 analyte concentrations. The $K_D$ values from each experiment were averaged and presented here with the standard error of the mean. Each SPR antibody-binding experiment was performed in triplicate. The signal value at each time point was normalized to either the negative control signal (in the antibody-binding experiments) or the duplex alone-binding signal (in the interaction inhibition experiments). The average of these values was plotted for each time point. The in vivo competition assay was performed in biological triplicate and each sample was processed independently. The protein bands after Western blot were quantified and statistically analyzed in GraphPad Prism (v 7.04, GraphPad Software, La Jolla, CA, USA) using unpaired two-tailed $t$-test.

**Reporting summary**. Further information on research design is available in the Nature Research Reporting Summary linked to this article.

## Data availability

The cross-linking mass spectrometry data sets generated and analyzed during the current study are available in the PeptideAtlas repository (http://www.peptideatlas.org/PASS/PASS01391). The PDB structures 2PBI and 6N9G were used for modeling. The source data for Figs. 4d, f and 6c are available as Supplementary Data 2–4.

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

## Acknowledgements

The authors thank the members of the Metabolic Diseases Branch, NIDDK for many helpful discussions and suggestions. The authors also thank Dr. Yong Chen from the NHLBI Proteomics Core for help with sample processing and data analysis. The Intramural Research Program of the National Institute of Diabetes and Digestive and Kidney Diseases (ZIA DK043304-24) supported this research.

## Author contributions

P.R.A., J.-H.Z., and W.F.S. conceived the idea, designed the experiments and prepared the manuscript. P.R.A. and J.-H.Z. performed the experiments. C.M.K. and S.A.H.

performed molecular dynamic simulations. C.M.K., M.P., and N.G.L. collected protein-modeling data. G.W. and M.G. performed mass spectrometry injections and analyzed cross-linking mass spectrometry data. All authors reviewed and edited the manuscript.

## Additional information

**Competing interests:** The authors declare no competing interests.

