## [Peer Review File · Communications Biology]

Reviewers' comments:

Reviewer #1 (Remarks to the Author):

Adikaram et al. use a combination of cross-linking mass spectrometry and computational modeling/docking to study the interaction of complexes involving the protein R7BP and its interactors. Distance restraints derived from XL-MS form the basis for a refined modeling step to generate models of R7BP itself and ternary complexes with R7BP, different forms of its binding partner RGS and Gbeta5. The models are interpreted in connection with SPR data from the four complexes.

Based on the suggested complex organization, antibodies were designed that were found to inhibit the interaction between R7BP and RGS/Gbeta5, and an N- and C-terminally truncated version of R7BP was designed and validated to act as a dominant negative inhibitor in cell culture experiments.

Taken together, the authors demonstrate that XL-MS can assist computational methods to generate models of individual proteins and protein complexes. While this does not constitute a completely novel approach per se, what differentiates this study from comparable work is that the proposed interactions were validated using different methods (SPR, generation of antibodies, use of truncations for in cell experiments). While I find the work of general interest, the manuscript needs some revisions before being acceptable for publication:

1. Presentation of cross-linking data

Although the experimental procedures are mostly presented in sufficient detail, there is only very little information about the identified cross-links / cross-linked peptides as part of some figures (see also my comments related to figure design below). More information about the identifications needs to be provided. I suggest to prepare a table that summarizes all the identified cross-links from all the different experiments and that contains a minimum level of information such as m/z ratios, charge state, mass error, identification scores and any other data that is useful to assess the quality of the matches. Moreover, it is not clear whether different cross-linker-to-protein ratios were used in all cases, whether/how the results were combined to the final list of cross-links and how much the overlap between different conditions was.

Assessing false discovery rates for small data sets where only few cross-links are identified is not straightforward. For example, if only five cross-links are identified in a particular experiment, it is completely unrealistic to specify a 1% FDR (irrespective of what the software claims), because even a single false positive would drive the FDR to >10%.

Deposition of the mass spectrometry data in a proteomics data repository is highly encouraged.

2. Use of XL-derived restraints for modeling

The authors specify a distance restraint of "10-20 Å" (lines 142, 528). Can the authors be more specific about how the restraints were defined in I-TASSER, as alpha-carbon distances, side-chain distances, etc.? 20 Å would be an unrealistically low upper bound of distance for Calphas.

Similarly, the procedure for how "best oriented models" (lines 531-532) were selected after ClusPro docking needs a more detailed

explanation.

In general, because the number of restraints was quite low, how did the authors make sure that the models were not biased by any one of the few restraints, given possible difficulties of accurate FDR estimates? From the presentation of the XL data in the figures, it is difficult to judge where the contacts are actually located.

3. Figure design

Almost all of the main text figures are heavily overloaded with many panels. While I understand the intention of the authors to present as much data as possible, some panels are very hard to decipher, especially those depicting SPR data. The figures should be revised, for example by splitting them into multiple figures. Some information could be moved to the Supporting Information, for example SPR curves if Kd data is already shown in bar graphs.

4. References

Some references, especially the more recent ones from 2018, are incomplete.

Reviewer #2 (Remarks to the Author):

This manuscript reports results of some useful and potentially informative experiments involving the R7-RGS protein docking adaptor protein R7BP: Lysyl residue cross-linking results for R7bp monitored by mass spectrometry, surface plasmon resonance measurements of binding of R7bp to different R7-RGS protein complexes with Gbeta5, similar binding measurements with antibodies recognizing different R7bp epitopes, and experiments showing blocking of R7bp complex formation by antibodies and by a dominant-negative construct. It also contains molecular modeling results of rather dubious quality and utility.

The experimental work and even the modeling are worth publishing; however, the claims that these results demonstrate some general sort of new structural strategy that can replace well-established structural methods (of which cross-linking and modeling are examples) are not to be taken seriously.

Logically, the presentation should be of the key experimental input, the cross-linking results, followed by their use as constraints for modeling. However, little detail is provided on how the constraints were used in the modeling process. Proximity of 10-20 Angstrom is quite a weak constraint, so it is important to report how the model was changed when generated with and without these constraints. The two models should be superimposed and the rmsd deviations of the C α carbons should be reported. Overall, the presentation as, (to paraphrase) "we generated a model and then tested it with experiments," does not really make sense, and is missing the acknowledgment that the only things that make the model remotely plausible are the two kinds of experimental data that went into it: the crystal structure of the syntaxin used as the homology template and the cross-linking data.

The modelling of separated R7 RGS proteins and Gbeta5 structures would be irrelevant except that it suggests the modeling approach is not very good. It apparently generated structures that are virtually identical to the two subunits of the RGS9-Gbeta5 dimer, which only exists in dimer form. Neither of these proteins is stable without the other or can even fold properly on its own. Obviously the modeling is not probing a very wide range of conformation space if it produces these separate subunits as stable structures. Therefore, there is little reason to trust its output for an unknown structure, except for the fact that at least R7bp is stable on its own and likely has a fold similar to syntaxins, as the

model predicts (and other modeling software also predicts) and as can be surmised without any modeling from sequence similarity. Also, the protein-protein docking is not described in sufficient detail: how much flexibility was allowed for both partners? How were the distance constraints used, and how did they affect the docking results? How many of the ranked poses were consistent with the distance constraints, etc.?

The antibody purification scheme described could only be effective if each peptide contains a Strep-tag (or Strep-tag II) sequence, but the Materials and Methods section does not mention this key detail. It is hard to understand why they isolated antibodies from serum instead of panning clones from a library derived from B-cell mRNA, which would have provided an unlimited resource for producing the antibodies, and allow selection of those with highest affinity and specificity.

There are no data presented on antibody purity, specificity or affinity other than the SPR measurements which yield no information on the epitopes recognized. Epitope mapping for a peptide-derived antibody is relatively easy with tools the authors have in hand. Unfortunately, the interpretation of the antibody binding data relies heavily on the assumption that their binding is strictly through the sequences used for immunization. Also, the peptides were rather long, so it is likely that only a sub-sequence is really involved in binding- however with polyclonals a mixture of many specificities is possible, further clouding the interpretation. In order to interpret the SPR signals for the antibody binding we need to know the concentrations of the antibodies used. In Fig. 4, the y-axis should not be labeled, "relative antibody binding affinity;" it is actually a time course of SPR signal and cannot even be used to estimate the affinity unless the antibody concentrations are known.

Minor technical point: sufficient information to identify the exact sequences used in all the expression constructs should be provided. For example, presumably for RGS9 experiments full-length RGS9-2, the physiological binding partner of R7bp, was used, but the way the manuscript is written it could also be the truncated version of RGS9-1 used in the crystal structure.

COMMSBIO-19-0255-T Reviewers' comments and **replies**:

Reviewer#1

1. Reviewer comment:

Although the experimental procedures are mostly presented in sufficient detail, there is only very little information about the identified cross-links / cross-linked peptides as part of some figures (see also my comments related to figure design below). More information about the identifications needs to be provided. I suggest to prepare a table that summarizes all the identified cross-links from all the different experiments and that contains a minimum level of information such as m/z ratios, charge state, mass error, identification scores and any other data that is useful to assess the quality of the matches. Moreover, it is not clear whether different cross-linker-to-protein ratios were used in all cases, whether/how the results were combined to the final list of cross-links and how much the overlap between different conditions was.

Reply:

We fully agree with the reviewer's comments and accordingly the relevant figures (Figs. 1C-D, 2G-H, 5B, and 6A) were redesigned to highlight the cross-linked lysines and the interactions between them. Details about DSSO concentrations used in each experiment, the cross-linked peptides obtained and the frequency with which each interaction was identified are detailed in the newly created table, Table S1. The Materials and Methods section now provides details on how the results were combined (lines 516-518). Table S1 also provides information on the overlap between different conditions.

Furthermore, details on m/z ratios, charge state, mass error, identification scores and other useful data were uploaded onto the proteomics data repository PeptideAtlas.org and will be available once the article is published (see lines 518-520).

2. Reviewer comment:

Assessing false discovery rates for small data sets where only few cross-links are identified is not straightforward. For example, if only five cross-links are identified in a particular experiment, it is completely unrealistic to specify a 1% FDR (irrespective of what the software claims), because

even a single false positive would drive the FDR to >10%.

Reply:

We appreciate this comment by the reviewer. Information on false discovery rate and our confidence in the results is provided in the revised Materials and Methods section under “Cross-linking mass spectrometry” (lines 507-515).

3. Reviewer comment:

Deposition of the mass spectrometry data in a proteomics data repository is highly encouraged.

Reply:

This is a very good suggestion, and all mass spectrometry data has now been deposited in the proteomics data repository PeptideAtlas.org.

4. Reviewer comment:

The authors specify a distance restraint of "10-20 Å" (lines 142, 528). Can the authors be more specific about how the restraints were defined in I-TASSER, as alpha-carbon distances, side-chain distances, etc.? 20 Å would be an unrealistically low upper bound of distance for Calphas.

Reply:

More specific details about the 10-20 Å distance restraints have been added to the Materials and Methods section under “Protein modeling” (lines 568-570).

5. Reviewer comment:

Similarly, the procedure for how "best oriented models" (lines 531-532) were selected after ClusPro docking needs a more detailed explanation.

Reply:

The procedure for how we performed ClusPro docking and selected the model to use has now been added to the Materials and Methods under “protein modeling” (lines 576-586).

6. Reviewer comment:

In general, because the number of restraints was quite low, how did the authors made sure that the models were not biased by any one of the few restraints, given possible difficulties of accurate FDR estimates? From the presentation of the XL data in the figures, it is difficult to judge where the contacts are actually located.

Reply:

Concerns about FDR have been addressed in the revised Materials and Methods section (lines 507-515). All models were created in an unsupervised fashion by the program. We have re-drawn the figures to show where the contacts are located (Figs. 1C-D, 2G-H, 5B, 6A).

7. Reviewer comment:

Almost all of the main text figures are heavily overloaded with many panels. While I understand the intention of the authors to present as much data as possible, some panels are very hard to decipher, especially those depicting SPR data. The figures should be revised, for example by splitting them into multiple figures. Some information could be moved to the Supporting Information, for example SPR curves if Kd data is already shown in bar graphs.

Reply:

As per the reviewer's helpful suggestion, we have rearranged the figures and moved some data to the Supplemental section.

8. Reviewer comment:

Some references, especially the more recent ones from 2018, are incomplete.

Reply:

The literature references were carefully reviewed and are now complete.

Reviewer #2 (Remarks to the Author):

This manuscript reports results of some useful and potentially informative experiments involving the R7-RGS protein docking adaptor protein R7BP: Lysyl residue cross-linking results for R7bp monitored by mass spectrometry, surface plasmon resonance measurements of binding of R7bp to different R7-RGS protein complexes with Gbeta5, similar binding measurements with antibodies recognizing different R7pb epitopes, and experiments showing blocking of R7bp complex formation by antibodies and by a dominant-negative construct. It also contains molecular modeling results of rather dubious quality and utility.

1. Reviewer comment:

The experimental work and even the modeling are worth publishing; however, the claims that these results demonstrate some general sort of new structural strategy that can replace well-established structural methods (of which cross-linking and modeling are examples) are not to be taken seriously.

Reply:

We did not intend to suggest that the reported methods are in any way meant to replace well-established structural methods. Instead we wanted to illustrate how these methods could be combined to shed light on, at least to some extent, the structural aspects of the proteins that might be difficult to solve by either existing method alone. The limited structural information could make it possible, in principle, to advance drug design against these protein targets. We have re-phrased the text to make it more clear so that the readers will not be misled.

2. Reviewer comment:

Logically, the presentation should be of the key experimental input, the cross-linking results, followed by their use as constraints for modeling. However, little detail is provided on how the constraints were used in the modeling process. Proximity of 10-20 Angstrom is quite a weak constraint, so it is important to report how the model was changed when generated with and without these constraints.

Reply:

We appreciate the point being made by the reviewer. Additional detail on how constraints were used in the modeling process, including how the 10-20 Å distance restraint was applied, as well as the changes in the model generated with or without constraints are described in the revised Materials and Methods section “Protein modeling” (lines 568-570, 576- 586).

3. Reviewer comment:

The two models should be superimposed and the rmsd deviations of the Calpha carbons should be reported.

Reply:

In response to the reviewer’s suggestion, the superimposed R7BP-FL models with and without restraints are now shown in Fig. 1E with RMSD values. The superimposed R7BPΔNC models with and without restraints are shown in Figs. 5C and S14A.

4. Reviewer comment:

Overall, the presentation as, (to paraphrase) “we generated a model and then tested it with experiments,” does not really make sense, and is missing the acknowledgment that the only things that make the model remotely plausible are the two kinds of experimental data that went into it: the crystal structure of the syntaxin used as the homology template and the cross-linking data. The modelling of separated R7 RGS proteins and Gbeta5 structures would be irrelevant except that it suggests the modeling approach is not very good. It apparently generated structures that are virtually identical to the two subunits of the RGS9-Gbeta5 dimer, which only exists in dimer form. Neither of these proteins is stable without the other or can even fold properly on its own. Obviously the modeling is not probing a very wide range of conformation space if it produces these separate subunits as stable structures. Therefore, there is little reason to trust its output for an unknown structure, except for the fact that at least R7bp is stable on its own and likely has a fold similar to syntaxins, as the model predicts (and other modeling software also predicts) and as can be surmised without any modeling from sequence similarity.

Reply:

In order to address the reviewer's concerns, we have re-phrased the text to better explain the rationale and logic for each study. Accordingly, the method by which I-TASSER models proteins is described in more detail in the Materials and Methods section under "protein modeling" (lines 562-565). The human RGS7, RGS9 and Gβ5 models that were generated similar to the crystal structures of their mouse counterparts, respectively, were used to demonstrate the utility of this modeling method. We agree with the reviewer that RGS9 and Gβ5 proteins are not stable as monomers *in vivo*. The methods reported here can only shed light on the portion of a protein that shares homology with a structure or structures in the database and, after integration with the experimentally-derived cross-linking data, it is hoped that at least in some cases the partial structural information can be used for drug design.

5. Reviewer comment:

Also, the protein-protein docking is not described in sufficient detail: how much flexibility was allowed for both partners? How were the distance constraints used, and how did they affect the docking results? How many of the ranked poses were consistent with the distance constraints, etc.?

Reply:

The reviewer's concerns are on point, and more details on ClusPro docking, how the distance constraints were used, and how they affected the docking results are now described in the Materials and Methods section "protein modeling" (lines 576-586).

6. Reviewer comment:

The antibody purification scheme described could only be effective if each peptide contains as Strep-tag (or Strep-tag II) sequence, but the Materials and Methods section does not mention this key detail.

Reply:

We appreciate the reviewer catching this oversight. Yes, the reviewer is correct that all peptides contained the same Twin Strep tag sequence. The presence of the Twin Strep tag sequence was now added to the Materials and Methods section under “antibody purification” (lines 438-440).

7. Reviewer comment:

It is hard to understand why they isolated antibodies from serum instead of panning clones from a library derived from B-cell mRNA, which would have provided an unlimited resource for producing the antibodies, and allow selection of those with highest affinity and specificity.

Reply:

We respect the scientific strategy proposed by the reviewer. We chose however to isolate and purify antibodies using the method described in the paper because of its cost and time effectiveness. Another advantage of this method is that corresponding single-domain nanobodies could be isolated if needed down the road. The reasoning is now also provided in the Materials and Methods section “antibody purification” (lines 437-438).

8. Reviewer comment:

There are no data presented on antibody purity, specificity or affinity other than the SPR measurements which yield no information on the epitopes recognized. Epitope mapping for a peptide-derived antibody is relatively easy with tools the authors have in hand. Unfortunately, the interpretation of the antibody binding data relies heavily on the assumption that their binding is strictly through the sequences used for immunization. Also, the peptides were rather long, so it is likely that only a sub-sequence is really involved in binding- however with polyclonals a mixture of many specificities is possible, further clouding the interpretation. In order to interpret the SPR signals for the antibody binding we need to know the concentrations of the antibodies used.

Reply:

According to the reviewer’s suggestions, the data on antibody purity and concentration determination are now provided in Fig. S13A. Fig. S13B details antibody specificity to

R7BP and not the RGS7/G β 5 duplex. Antibody affinity to R7BP is shown in Fig. 4D and details on other antibodies not shown in the figure are described in the Results section under “development of antibody inhibitors of itch” (lines 255-259).

We are aware that in these experiments we used polyclonal antibodies as a first step for screening purposes. In the second phase of study we plan to do epitope mapping and isolate corresponding monoclonal antibodies, which is explained in the Materials and Methods section “antibody purification” (lines 458-460). A concentration of 20 μ g/mL antibody was used in the SPR experiments, a detail now included in the Materials and Methods section “SPR spectroscopy” (line 533).

9. Reviewer comment:

In Fig. 4, the y-axis should not be labeled, “relative antibody binding affinity;” it is actually a time course of SPR signal and cannot even be used to estimate the affinity unless the antibody concentrations are known.

Reply:

The axis of the graph in Fig. 4D has been changed according to the reviewer’s suggestions.

10. Reviewer comment:

Minor technical point: sufficient information to identify the exact sequences used in all the expression constructs should be provided. For example, presumably for RGS9 experiments full-length RGS9-2, the physiological binding partner of R7bp, was used, but the way the manuscript is written it could also be the truncated version of RGS9-1 used in the crystal structure.

Reply:

As helpfully suggested by the reviewer, the details of the GNB5 short form and RGS9-2 constructs used in the experiments have been added to the Materials and Methods section under “DNA constructs” (line 390).

Reviewer #3

1. Reviewer comment:

The mass spectrometry data could be better represented and utilized. The authors show the peptides but don't give any quantitative information, such as replicates, and abundance of peptides. This is especially concerning as they compare slight changes in the number of crosslinks, which seems risky to compare if there is only one replicate as mass spectrometry is hard to replicate and there are often false positives. Furthermore, the authors never map the crosslinks on the structure nor show their distances, which would be especially helpful in knowing if they help to confirm their models. In general, there are no strong assessments for their models or crosslinks. The authors treat their model as equal to a crystal structure, however with a crystal structure there would be more statistical measures to validate the model.

Reply:

With this comment, the reviewer has made several very helpful points for the better presentation of the data. Accordingly, the mass spectrometry data has been rearranged and better represented. Table S1 presents detailed data on XL-MS experiments, DSSO concentrations, peptides obtained and frequency of each cross-linked peptide pair. More details of experimental results have been uploaded to PeptideAtlas.org. We have also provided details on how we addressed false positives in the Materials and Methods section "XL-MS" (lines 507-515). Figs. 1C-D, 2G-H, 5B, 6A show more details of the crosslinks on the structure and Tables S2-3, S5-9 provide distances between each cross-linked lysine.

2. Reviewer comment:

One concern is Figure S8, which shows electrostatic potential as a scale from hydrophilic to hydrophobic. There is no information in the legend on how this figure was generated, though typically in PyMOL it would be with the vacuum electrostatics command, which would then give a scale from negative (red) to positive (blue). Can the authors explain how they generated this figure with more detail and if the scale indeed represents hydrophobicity or charge? The authors explain that two sites could be interacting as both being hydrophobic, yet they would repel if both are positive (blue). The sequence itself doesn't look particularly hydrophobic or

basic by eye, but it is hard to know without seeing the exact residues mapped on the surface. Please check the sequence and the scale, and if it is the correct sequence but based on charge, is there still a good reason to suggest why R7BP would specifically interact with RGS7 over RGS6? Also, APBS is a better plug-in to generate electrostatics in PyMOL (instead of vacuum electrostatics).

Reply:

In response to this suggestion, we re-generated Fig. S9 (previously Fig. S8) using the APBS plugin as recommended by the reviewer. This figure depicts the electrostatic surface potentials of RGS6, RGS7 and R7BP. Since we could not calculate a charge for the interaction surface of R7BP, we could not definitely hypothesize that the difference in R7BP binding affinity between RGS6 and RGS7 is due to electrostatics. It may be due to differences in hydrophobicity on the interaction interfaces, as depicted in Fig. S10, or a combination of both.

3. Reviewer comment:

The rationale behind the dominant negative is never explained, and the authors hardly mention the palmitoylation site, which is presumably the reason for the dominant negative effect.

Reply:

We appreciate this comment. The rationale for the dominant negative construct is now explained in the Results section under “dominant negative as potential inhibitors of R7BP” (lines 268-275) and underscores the essential role of the palmitoylation site at its C terminus. Since both the N and C terminus of R7BP lend instability to the protein, we hypothesized that the R7BP Δ NC construct would be more stable and bind to R7-RGS proteins with higher affinity. We also mention the role of the palmitoylation site in the Results section (lines 145-148) and that we mutated this site in the Materials and Methods section (lines 391-392). In all our experiments the palmitoylation site is absent, so it has no effect on the binding affinities we measured.

4. Reviewer comment:

In their analysis, other measures could help their modeling, such as the location of the membrane-anchored sites and if they are on the same face of their model? Thus could they propose how this complex would interact with membranes?

Reply:

The membrane-anchored site (palmitoylation site) is located at AA 252-253 (Materials and Methods, lines 391-392) which is at the C-terminus. The focus of this paper was to study the interactions between R7BP and R7-RGS proteins, and as a result we did not concentrate on its interaction with the membrane. A recent paper that explores this interaction in more depth is: PMID: 26811338.

5. Reviewer comment:

Did they analyze the buried surface area for the different complexes they modeled, and do they line up with the differences in affinity they observe?

Reply:

We analyzed the buried surface area for all R7-RGS complexes as shown in Table S4. We found no correlation between buried surface area and binding affinity. This information has been inserted into the corresponding portion of the Results section (lines 186-189).

6. Reviewer comment:

Is ab38 specific to RGS7/G β 5? It seems like it should be based on the model?

Reply:

Ab38 only specifically interacts with R7BP and does not interact with RGS7/G β 5, as shown in Fig. S13B.

7. Reviewer comment:

Mass spectrometry is not a “standard benchtop operation”. Please take that out of the abstract.

Reply:

We have removed this phrase from the abstract.

8. Reviewer comment:

Significant figures overall, like in the SPR legends, you do not know your concentration down to the hundredth of nM (i.e. 106.92, and make them consistent). Also in Table 1 it would be easier to read them in nM and then take out the extra sig figs – would read better as 2.2 ± 0.8 nM.

Reply:

We appreciate these points and have made these changes to Table 1 and the SPR figures according to the reviewer's suggestions.

9. Reviewer comment:

Why exclude G β 5 from figure 3? Could make it somewhat transparent but it would help to see it.

Reply:

The cartoon structure of G β 5 was added to all figures in Fig. 3B according to the reviewer's suggestions.

10. Reviewer comment:

What are the RMSD values for your models versus the crystal structures for the RGS7 and RGS9 complexes?

Reply:

We calculated the RMSD values for RGS7/G β 5 and RGS9/G β 5 models and crystal structures which are now shown in Fig. S4B.

Reviewers' comments:

Reviewer #1 (Remarks to the Author):

In this revised version, Adikaram et al. have made substantial changes to their manuscript. Most importantly, they provided additional experimental details and more information about the cross-link identifications and their use in the modeling procedures. This way, it is easier to follow how the authors obtained their results. In addition, the text was revised in response to the referees' comments, and the figures were revised in order to make them easier to follow.

The following references are still incomplete and lack volume, page number or article number information: 18, 21, 23, 35, 46, 56.

Apart from that, I do not have any further recommendations for this manuscript.

Reviewer #2 (Remarks to the Author):

The extensive revisions have largely addressed the concerns raised in the previous round of review. There is something seriously wrong with the far-right pair of structures in Fig. 2E. It looks like the structural alignment was not done properly, as none of the helices are aligned between the two structures.

Reviewer #3 (Remarks to the Author):

The authors have made many of the changes and greatly improved the manuscript as suggested by all the reviewers. The addition of Table S1, more information on the distances they measured as well as showing the crosslinks in figures is all helpful. However, there is still some confusion about the number of replicates/experiments for the crosslinking. They show two different ratios as two different experiments. And elsewhere in line 515 they mention replicates. But still it is not clear if each of these different ratios were performed more than once? It seems so, but it would be helpful to spell this out somewhere. In Table S1, it would help to say the exact residue numbers that were crosslinked with each peptide, and then how often each of those crosslinks show up in different replicates, and between the two experiments. Ideally, the table (and modeling) would only show crosslinks that were consistent between multiple replicates and/or experiments.

Of other minor note, the coloring used in Table S2 is not clear. And it would help to have the crosslinked residue numbers listed as a separate column.

Reviewers' comments re: COMMSBIO-19-0255-A:

Reviewer #1 (Remarks to the Author):

In this revised version, Adikaram et al. have made substantial changes to their manuscript. Most importantly, they provided additional experimental details and more information about the cross-link identifications and their use in the modeling procedures. This way, it is easier to follow how the authors obtained their results. In addition, the text was revised in response to the referees' comments, and the figures were revised in order to make them easier to follow.

The following references are still incomplete and lack volume, page number or article number information: 18, 21, 23, 35, 46, 56.

Apart from that, I do not have any further recommendations for this manuscript.

Reply:

Thank you for the positive feedback. We appreciate the remarks on the references and have added the volume and page number information for each of the references described.

Reviewer #2 (Remarks to the Author):

The extensive revisions have largely addressed the concerns raised in the previous round of review. There is something seriously wrong with the far-right pair of structures in Fig. 2E. It looks like the structural alignment was not done properly, as none of the helices are aligned between the two structures.

Reply:

Thank you for the feedback. As we examined the figures again we thought it likely that you meant Fig. 1E instead of Fig. 2E. The stated discrepancy was due to the alignment conditions either with or without refinements. We have added a new figure, Fig. 1F, with a second alignment and described more explicitly the conditions for performing the alignments.

Reviewer #3 (Remarks to the Author):

The authors have made many of the changes and greatly improved the manuscript as suggested by all the reviewers. The addition of Table S1, more information on the distances they measured as well as showing the crosslinks in figures is all helpful. However, there is still some confusion about the number of replicates/experiments for the crosslinking. They show two different ratios as two different experiments. And elsewhere in line 515 they mention replicates. But still it is not clear if each of these different ratios were performed more than once? It seems so, but it would be helpful to spell this out somewhere. In Table S1, it would help to say the exact residue numbers that were crosslinked with each peptide, and then how often each of those crosslinks show up in different replicates, and between the two experiments. Ideally, the table (and modeling) would only show crosslinks that were consistent between multiple replicates and/or experiments.

Of other minor note, the coloring used in Table S2 is not clear. And it would help to have the crosslinked residue numbers listed as a separate column.

Reply:

Thank you for your comments and we agree that the descriptions about the experiments and replicates were somewhat confusing. For better clarification, we have added a summary table (Table S1) that lists the number of experiments we performed for each of the proteins. We also modified the original Table S1 (now renamed as Table S2, please see details below). Since we have high confidence in the XL peptides we identified due to stringent FDR conditions (described in the Materials and Methods section), we were more interested in identifying a maximum number of different XL peptides, rather than repeating the experiment under identical conditions, to obtain more structural information. In each case, we performed the XL experiments for different protein complexes at different ratios between DSSO and proteins.

As stated above in the new Table S2, we have added residue numbers to XL peptides and added information on XL residues, as suggested. There is one column that specifies the frequency that a XL peptide appears in the experiment (PSM) and have added a column that shows the total number of times that the XL peptide appears in all the experiments performed. This indicates the reproducibility of our XL peptide results across different experiments and conditions.

As suggested, we have added the cross-linked residue numbers in a new column. The coloring codes in the supplemental sections are detailed in the corresponding Figure Legends.

REVIEWERS' COMMENTS:

Reviewer #2 (Remarks to the Author):

The remaining serious problems have now been addressed.

Reviewer #3 (Remarks to the Author):

Tables S1 and S2 are much more clear and I now understand the different experiments, and ratios, etc. I think the readers will appreciate this format. I have no further comments.